# AsyncTool: Evaluating the Asynchronous Function Calling Capability under Multi-Task Scenarios

## Abstract

Large language models (LLMs) based agents have demonstrated strong proficiency in leveraging external tools to address complex problems. However, existing evaluations largely overlook the temporal dimension of tool invocation, particularly the practical impact of inherent tool response latency, and they are typically confined to single-task scenarios. In realistic applications, tasks often need to be executed in parallel, and overall efficiency critically depends on the ability to utilize idle time during tool response delays. We denote this capability as asynchronous tool calling. To address the lack of evaluation in this area, we propose AsyncTool, which, to the best of our knowledge, is the first benchmark specifically aimed at assessing the asynchronous multitasking abilities of LLM-based agents within interactive tool-use contexts. AsyncTool consists of composite tasks with intra-task step dependencies that must be executed concurrently while incorporating realistic tool response delays. Through a hybrid data evolution strategy, we construct a diverse and representative asynchronous multitasking dataset that covers multiple scenarios and exhibits a wide range of tool use patterns. We further assess performance from three levels, namely Step Level, Sub-Task Level, and Task Level, covering perspectives from fine-grained to coarse-grained. Extensive experiments on AsyncTool show that even state of the art models experience notable performance degradation when confronted with complex asynchronous workflows. Our analysis identifies the main failure modes of current tool agents and provides practical guidelines for designing future systems with stronger temporal reasoning and coordination capabilities.

## 1 Introduction

With recent advancements in large language models (LLMs) in their ability to follow instructions and understand contextual information, LLM-based agents have become increasingly proficient in tool use (OpenAI, 2025b; Comanici et al., 2025; Anthropic, 2025; Yang et al., 2025; Kimi et al., 2025; Zeng et al., 2025; Chen et al., 2025; Wang et al., 2025; Chen et al., 2024b). This capability enables them to handle more sophisticated, multi-step tasks and to achieve strong performance across diverse tool-use scenarios (Liu et al., 2023).

However, real-world environments are often more complex, frequently requiring the concurrent execution of multiple tasks that may involve different tools. In practical settings, function calls usually incur latency, and executing tasks sequentially in a synchronous manner fails to fully utilize idle time, thereby reducing overall efficiency. To better evaluate and enhance the agent's performance under such conditions, we introduce the concept of **Asynchronous Tool Call** into the interaction between the agent and the environment, where the agent should utilize these idle intervals to advance other tasks. Motivated by these gaps, we identify three critical observations: **(i) Inadequate evaluation of the agent's capability to complete multiple tasks in asynchronous scenarios.** Existing studies are typically restricted to single-task scenarios in which tools operate in an immediate response manner (Zhuang et al., 2023; Ruan et al., 2023; Xu et al., 2023; Guo et al., 2024; Qin et al., 2023; Ye et al., 2024), overlooking the evaluation for multiple tasks in asynchronous scenarios. **(ii) Lack of alignment with real-world conditions in interactive environments involving real-time tool calls.** Existing asynchronous planning benchmarks do not operate within interactive environments, which is inconsistent with real-world scenarios involving real-time tool calls (Lin et al., 2024). **(iii)**

Table 1: **Comparison between ASYNCTOOL and existing benchmarks.**

| Benchmark | Async Executor | Function Call | Multi Task | Multi Step | Cross Scenario |
|---|---|---|---|---|---|
| τ-bench (Yao et al., 2024) | ✗ | ✓ | ✗ | ✗ | ✗ |
| BFCL v3 (Yan et al., 2024) | ✗ | ✓ | ✗ | ✓ | ✗ |
| NestFul (Basu et al., 2024) | ✗ | ✓ | ✗ | ✓ | ✗ |
| TimeArena (Zhang et al., 2024a) | ✓ | ✗ | ✓ | ✓ | ✗ |
| Cˆ3-Bench (Yu et al., 2025a) | ✗ | ✓ | ✓ | ✓ | ✗ |
| Robotouille (Gonzalez-Pumariega et al., 2025) | ✓ | ✗ | ✗ | ✓ | ✗ |
| **ASYNCTOOL (Ours)** | ✓ | ✓ | ✓ | ✓ | ✓ |

**Insufficient metrics and standardized protocols specific to concurrent tasks with delayed and out-of-order tool feedback.** Traditional benchmarks involving time delays do not cover tool-using tasks and cannot be transferred to agentic tasks (Zhang et al., 2024a; Gonzalez-Pumariega et al., 2025).

Building upon these insights, we propose ASYNCTOOL, the first benchmark specifically designed to evaluate the ability of LLM-based agents to perform asynchronous tool calling in multi-task scenarios. (i) Our benchmark comprises combinations of multiple tasks, each containing intra-task step dependencies, while the tasks themselves run in parallel. This design allows a model to advance other tasks concurrently during the waiting periods for tool responses. (ii) To more accurately reflect real-world conditions, we assign realistic response times to each tool, integrate multiple tasks, and require the agent to progress them concurrently through asynchronous function calls. This framework provides a practical and realistic environment for assessing asynchronous task execution capabilities. Table 1 presents a comparison between ASYNCTOOL and existing benchmarks on tool calling and asynchronous execution. (iii) To better evaluate the asynchronous capabilities of models, we assess performance from three levels: Step Level, Sub-Task Level, and Task Level, thereby enabling our benchmark to provide a comprehensive evaluation ranging from fine-grained to coarse-grained perspectives. Through extensive experiments, we observe that while some models demonstrate effective asynchronous behavior, most LLMs still encounter considerable challenges in handling asynchronous function-call tasks. Our detailed analysis of the experimental results offers further insights into the observed failure modes, providing valuable guidance for future research.

The main contributions of our work are summarized as follows:

- We propose ASYNCTOOL, the first framework to evaluate an agent's ability to perform multiple tasks through asynchronous tool calling, by explicitly simulating real-world tool response delays.

- We construct a high-quality asynchronous multitasking dataset, derived from single-task data using a hybrid strategy for data evolution. These composite tasks vary in quantity and encompass various scenarios, with each task comprising multi-step tool-call trajectories and clear function dependencies. This design guarantees complexity and challenge while maintaining the validity of the tasks.

- Through extensive experiments, we provide a detailed characterization of the challenges faced by tool-calling agents in asynchronous multitasking, examine the key difficulty factors affecting their performance, and conduct quantitative analyses that offer actionable insights for future research.

## 2 ASYNCTOOL

Building upon the motivation introduced in Section 1, ASYNCTOOL is designed to realistically simulate tool response latency and evaluate an agent's asynchronous tool calling capabilities. This section first formalizes the interaction paradigm used in ASYNCTOOL, which models how agents manage multiple tasks concurrently under delayed tool feedback. We then describe the construction of the high-quality asynchronous multitasking dataset and present the evaluation protocol.

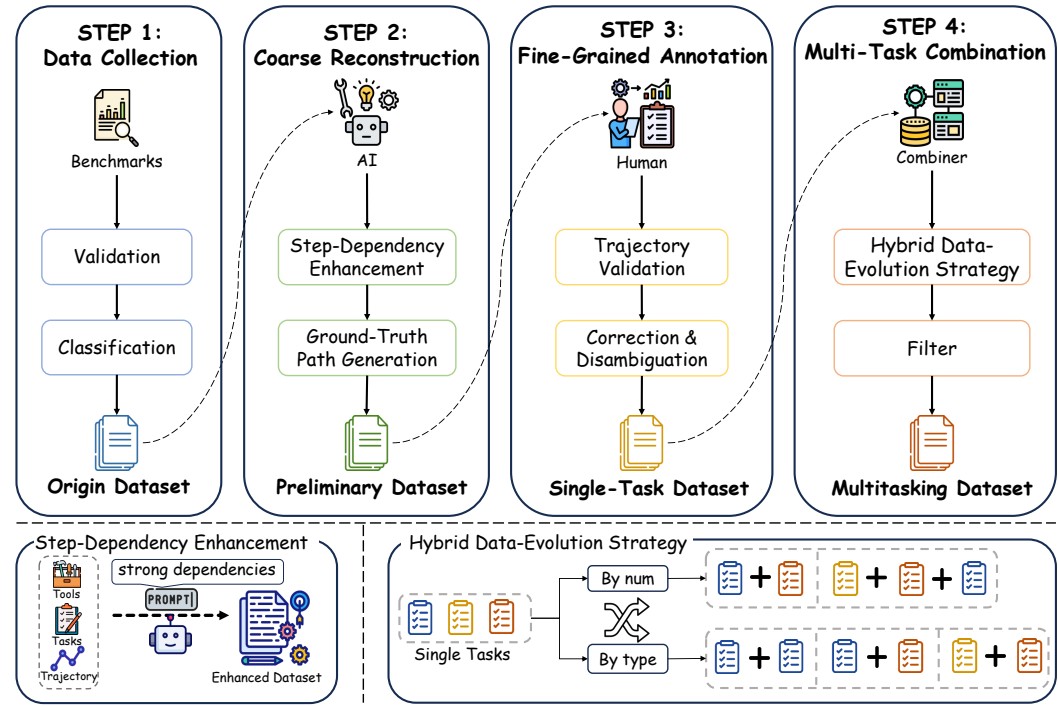

Figure 1: **Overview of the dataset construction process.** The pipeline starts by collecting raw data from tool-use benchmarks, categorizing it by scenario. Task-step dependencies are reinforced, and execution trajectories are reconstructed using Gemini 2.5 Pro, followed by manual verification for accuracy and determinism. Finally, data evolution occurs through hybrid strategies, with filtering producing the final multitasking dataset.

## 2.1 AGENT AS CONCURRENT TOOL-USING SYSTEM

While agents' ability to solve problems through tool use is well established, their practical effectiveness is often hindered by the non-negligible response latencies of tool calls in real-world scenarios. This raises the research question of whether agents can exploit such idle periods to make progress on other tasks. ASYNCTOOL is designed to address this challenge by simulating real-world scenarios to evaluate an agent's ability to asynchronously utilize tools and complete tasks. The framework explicitly models two key characteristics of real-world tool use: the tool response latency and the concurrent submission of multiple tasks. In ASYNCTOOL, the interaction pattern between the agent and tools differs fundamentally from that in conventional tool use settings: the results of all tool calls are not returned immediately but are intentionally delayed by a predefined waiting period. This design compels the model to advance other tasks during the idle time created by tool response latency. Consequently, delayed feedback and multitasking are essential components of ASYNCTOOL, distinguishing it from existing tool-use benchmarks.

For example, consider a scenario in which the agent receives two tasks, denoted as $task_1$ and $task_2$, requiring the function calls $\langle f_1, f_3, f_5 \rangle$ and $\langle f_2, f_4 \rangle$, respectively. The two tasks are independent of each other, while the function calls within each list must be executed sequentially due to dependency constraints. In this setting, the agent acts as the *Assistant*, while the execution system functions as the *Environment*. The *Assistant* first attempts to solve $task_1$ by calling $f_1$. After producing the formatted output for this call, the *Environment* immediately reports that $f_1$ has not yet returned a result, as tool executions are inherently non-instantaneous. At this point, the *Assistant* switches to $task_2$ and issues the call $f_2$, which similarly incurs its own waiting time. By the time the *Assistant* completes formatting the call for $f_2$, the *Environment* provides the return value for $f_1$, enabling the *Assistant* to resume execution of $task_1$. This alternating process continues until all tasks are completed.

The above example can be concretely realized within our framework, as illustrated in Figure 2. Conceptually, the framework models the large language model as a **concurrent system**, allowing it

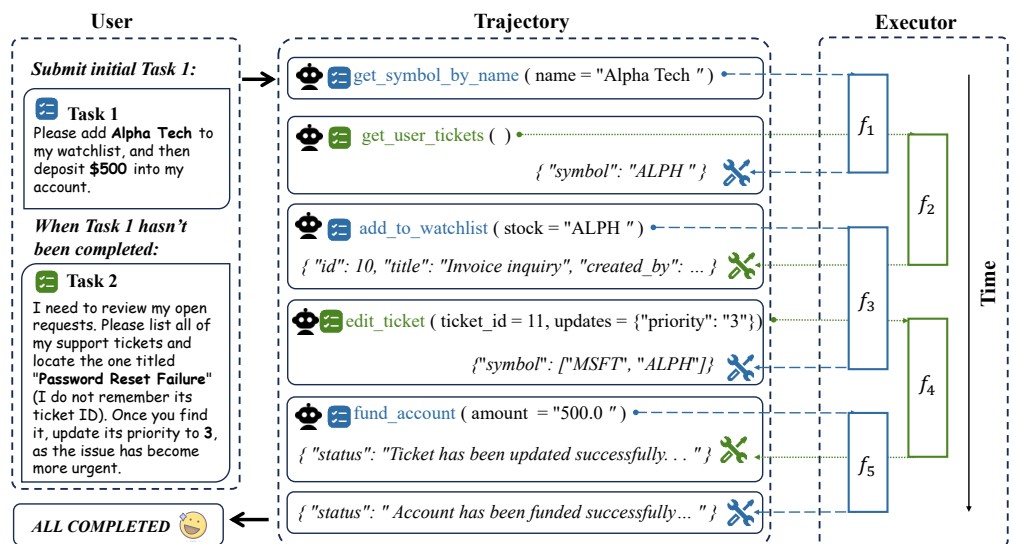

Figure 2: **An example of multi-task completion through asynchronous function calls.** In this scenario, the model receives tasks concurrently, and the executor simulates real-world latency by delaying outputs, which forces the model to interleave calls across tasks.

to interact with tools asynchronously and execute multiple tasks in parallel. The proposed benchmark subsequently evaluates the model's capability to schedule and coordinate these tasks effectively.

## 2.2 DATA CONSTRUCTION

The construction of ASYNCTOOL requires a high-quality multitasking dataset. To this end, we propose a data construction pipeline consisting of four main steps: **Data Collection** (§ 2.2.1), **Coarse Reconstruction** (§ 2.2.2), **Fine-Grained Annotation** (§ 2.2.3), and **Multi-Task Composition** (§ 2.2.4). The overview of the dataset construction process is illustrated in Figure 1.

### 2.2.1 DATA COLLECTION

We note that existing benchmarks have already compiled tool APIs derived from real-world scenarios and provide well-developed tool executors, task descriptions, and execution paths. To avoid reinventing the wheel, we leverage these resources as suitable sources of high-quality single-task data. In our work, we select two representative benchmarks, NESTFUL (Basu et al., 2024) and BFCLv3 (Yan et al., 2024). After programmatic verification, we categorize and organize their tools and tasks, resulting in each task being uniquely associated with a specific tool category. Through this process, we extract a total of 12 tools and 358 tasks with corresponding tool-call paths as **Origin Dataset**.

### 2.2.2 COARSE RECONSTRUCTION

To ensure both consistency and accuracy of evaluation, we expect the data to follow a fixed tool-call trajectory for each task, which eliminates uncertainties in the calling sequence (Figure 4). Following this principle, we reinforce the dependencies between task steps by refining the task descriptions. Furthermore, to reduce the cost of manual annotation, we leverage Gemini 2.5 Pro (Comanici et al., 2025) to reconstruct the original task descriptions. To reduce the cost of manual annotation, we leverage Gemini 2.5 Pro (Comanici et al., 2025) to reconstruct the original task descriptions. Specifically, we provide the model with the original task description, multi-step execution trajectory, and tool set (Hsieh et al., 2023) as essential inputs, and sample instances in batches. Using carefully designed few-shot prompts, the model is guided to reconstruct the task descriptions and to generate strictly ordered function-call trajectories that are consistent with them. The specific prompt can be found in the Appendix C.5. Owing to the strong comprehension and reasoning capabilities of Gemini 2.5 Pro, the model produces reconstructed data that largely satisfies our requirements. Nonetheless, a

small number of issues remain, such as incorrect function arguments or mismatches between the task description and execution order, which require further manual annotation.

### 2.2.3 FINE-GRAINED ANNOTATION

After refining the data with Gemini 2.5 Pro, we obtain a **Preliminary Dataset** that still contains several potential issues, some of which are severe enough to invalidate the entire tool-call trajectory. To address these issues, we design a fine-grained human annotation pipeline aimed at eliminating data errors introduced during model-based generation.

**Trajectory Validation.** We first ensure that every function call in a trajectory is valid. To this end, we manually verify the sequential execution results for each task trajectory. Through this process, we identified three recurring error patterns and applied targeted corrections: (1) misinterpretation of the initial task conditions, leading to errors in the first call, such as repeatedly executing cd to enter the current directory in file system tasks; (2) violation of dependency relations, i.e., failing to invoke prerequisite functions, for example skipping a preceding call that is closely tied to the current one; and (3) misunderstanding of tool functionalities, often manifested as filling arguments in unsupported formats.

**Correction and Disambiguation.** Once the validation step confirms that all trajectories are free of execution errors, our focus shifts to aligning tasks with their corresponding trajectories and eliminating ambiguities. First, we verify the consistency between each task and its trajectory, removing task descriptions or partial trajectories that cannot be matched. Second, we strictly enforce the order of function calls within each trajectory, correcting any incorrect sequences. Finally, we replace ambiguous descriptions with precise expressions wherever possible, ensuring that essential details (*e.g.*, location, time, and other key arguments) are explicitly included in the task description.

Following this pipeline, we conduct multiple rounds of verification on 358 tasks until no errors remain, ultimately producing a high-quality **Single-Task Dataset** comprising 358 validated instances, as summarized in Table 4.

### 2.2.4 MULTI-TASK COMPOSITION

In real-world settings, multitasking often spans multiple scenarios and involves complex tool combinations. To more realistically evaluate an agent's capability for asynchronous function calls in multi-task scenarios, we focus on two factors: task quantity and task type. We introduce a hybrid data-evolution strategy, enabling the creation of diverse multitask combinations. Specifically, for task quantity, we define two categories: dual-task and tri-task. For task type, we define two categories: within-class and cross-class. By combining these factors, we obtain four distinct multitask configurations, which are applied to our single-task dataset to generate a rich multitask dataset.

However, applying the hybrid strategy results in an excessively large number of task combinations. To manage this, we adopt a weighted random sampling method to select a fixed-size subset of multitask data. The final **Multitasking Dataset** comprises 712 instances, effectively capturing the diversity and complexity of real-world multitasking scenarios.

### 2.3 EVALUATION

In **ASYNCTOOL**, each task is defined as a set of $n$ subtasks $\{S_1, S_2, \ldots, S_n\}$. Each subtask $S_i$ is represented as a tuple $(I, Q, T, E)$, where $I$ is a unique identifier, $Q$ denotes the task query, $T$ specifies the list of available APIs, and $E$ denotes the hidden state corresponding to the subtask, which is not directly exposed to the assistant. The model's response must explicitly include $I$ to indicate which subtask is being executed. For each subtask, we extract its execution trajectory $\mathcal{T}_i$, defined as an ordered sequence of tool calls: $\mathcal{T}_i = \langle a_1, a_2, \ldots, a_k \rangle$, where each action $a_j$ is represented as a tuple $(tool, args)$. Once all subtasks are completed, we obtain the set $\{\mathcal{T}_1, \mathcal{T}_2, \ldots, \mathcal{T}_n\}$, which is then used to evaluate whether the model has successfully accomplished the overall task.

In asynchronous multi-task execution, the interactions between the assistant and external tools can become highly complex. To provide a more comprehensive evaluation of the assistant's performance under such conditions, we assess the results along three dimensions: Step Level, Sub-Task Level, and Task Level.

Table 2: **Main Results of ASYNCTOOL.** Func. and Param. mean matching the model's results with the ground truth to calculate the F1 score. Char. means path matching while Env. means matching multiple execution result environments. **Bold** indicates best overall performance, while underline denotes the best within the same group.

| Models | Step Level | | Sub-Task Level | | | Task Level | | Overall |
|---|---|---|---|---|---|---|---|---|
| | Func. | Param. | Char. | Env. | Acc. | Char. | Env. | |
| *Closed-Sourced Models* | | | | | | | | |
| Qwen-Max | 86.22 | 73.62 | 53.45 | 72.98 | 52.44 | 26.97 | 50.14 | 25.56 |
| Kimi-K2 | 96.14 | 80.46 | 59.61 | 76.06 | 56.79 | 28.23 | 51.69 | 24.44 |
| Gemini 2.5 Pro | 89.08 | 78.27 | 63.16 | 76.43 | 62.05 | 34.13 | 54.35 | 32.44 |
| GPT-5 | 92.21 | 80.11 | 62.13 | 78.34 | 60.67 | 33.15 | 58.43 | 31.32 |
| GPT-4o | 93.92 | 82.26 | 62.90 | 80.04 | 61.41 | 33.71 | 60.53 | 31.74 |
| GPT-4.1 | **96.22** | **84.08** | **67.89** | **82.91** | **67.14** | **39.04** | **64.89** | **38.06** |
| *Open-Sourced Models < 20B* | | | | | | | | |
| LLaMA-3.1-8B-Ins. | 78.29 | 43.69 | 12.95 | 41.08 | 12.47 | 1.26 | 14.61 | 1.26 |
| GLM-4-9B-chat | 74.72 | 59.70 | 29.67 | 55.94 | 28.93 | 7.30 | 25.56 | 6.88 |
| Qwen2.5-7B-Ins. | 82.40 | 65.01 | 28.66 | 55.15 | 26.38 | 6.74 | 25.84 | 6.04 |
| Qwen2.5-14B-Ins. | 81.32 | 70.22 | 50.21 | 65.29 | 46.28 | 22.61 | 38.20 | 18.82 |
| Qwen3-8B | 63.05 | 53.61 | 29.88 | 53.08 | 29.30 | 11.10 | 28.65 | 10.67 |
| Qwen3-14B | 85.02 | 72.67 | 49.62 | 69.27 | 47.19 | 20.51 | 44.66 | 18.82 |
| *Open-Sourced Models > 20B* | | | | | | | | |
| LLaMA-3.1-70B-Ins. | 89.60 | 47.10 | 18.47 | 43.26 | 17.83 | 3.37 | 16.43 | 2.81 |
| LLaMA-3.3-70B-Ins. | 73.00 | 40.32 | 21.92 | 44.37 | 20.54 | 6.18 | 18.26 | 5.34 |
| GLM-4-32B | 60.59 | 51.41 | 35.46 | 56.21 | 33.97 | 16.57 | 29.78 | 15.17 |
| Qwen3-32B | 79.95 | 70.37 | 48.14 | 67.46 | 46.71 | 20.22 | 41.43 | 19.10 |
| Qwen2.5-32B-Ins. | 94.24 | 81.73 | 61.41 | 74.42 | 56.48 | 30.62 | 49.72 | 24.86 |
| Qwen3-30B-A3B-Ins. | 94.29 | 80.03 | 54.14 | 72.72 | 53.03 | 22.33 | 47.47 | 21.49 |
| DeepSeek-V3.1-Terminus | 86.10 | 75.32 | 59.39 | 72.45 | 56.21 | 33.43 | 49.30 | 28.93 |

**Step Level.** Following the fine-grained evaluation methodology of Patil et al. (2023), we assess the agent's fundamental tool-calling capability, focusing on aspects such as call format, tool verification, and parameter verification. To quantify these results, we adopt the approach of Basu et al. (2024) and compute the F1 score separately for tool accuracy and parameter accuracy.

**Sub-task Level.** At this level, we define accuracy-based metrics to evaluate the agent's performance on individual subtasks. For each subtask, we compare the assistant's predicted trajectory $\mathcal{T}_i^{\mathrm{pred}}$ with the ground-truth trajectory $\mathcal{T}_i^{\mathrm{gt}}$ to determine whether the subtask is successfully completed, yielding the **trajectory-completion metric**. In addition, we assess the subtask's hidden state by comparing $E_i^{\mathrm{pred}}$ and $E_i^{\mathrm{gt}}$ to measure consistency, yielding the **environment-matching metric**. Finally, these two metrics are combined into an **overall subtask accuracy**, which evaluates the agent's final performance on subtask completion. Detailed calculation procedures are provided in Appendix A.1.

**Task Level.** As the final evaluation, the trajectory-completion and environment-consistency metrics at this level are counted only when all corresponding subtask-level metrics within a task are satisfied. These measurements provide an overall assessment of the agent's ability to complete the entire task. The final overall task accuracy is defined as the proportion of tasks for which both conditions are simultaneously met.

## 3 EXPERIMENT

### 3.1 EXPERIMENTAL SETUP

We evaluate 19 models on ASYNCTOOL, aiming to provide a comprehensive benchmark for assessing their capability of asynchronous tool calling under multi-task scenarios. Specifically, for closed-source models, we select four prominent models: Qwen-max (Team, 2024c) created by the Qwen Team,

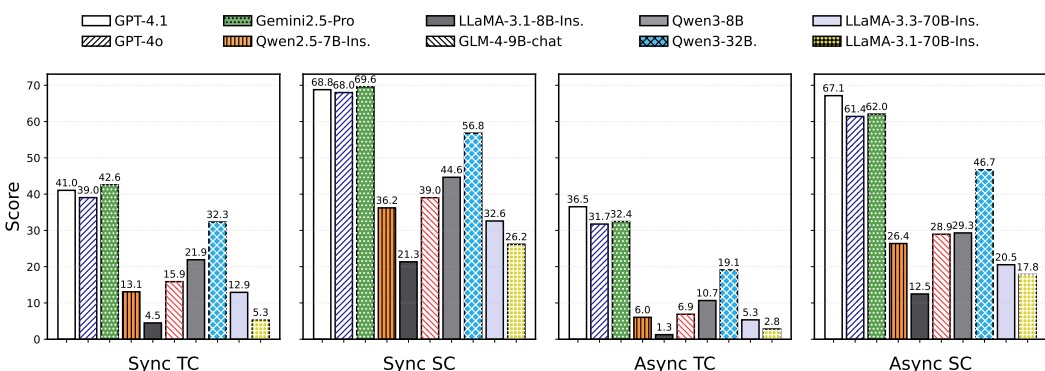

Figure 3: Scores of some models. TC is overall, and SC is Sub-Task according. As the model size reduces, the model score correspondingly declines. A higher subtask completion rate does not invariably result in a higher overall score.

Kimi k2 (Kimi et al., 2025) by Kimi Team, Gemini 2.5 Pro (Comanici et al., 2025) developed by Google, alongside GPT-4.1 (Achiam et al., 2023), GPT-4o (Hurst et al., 2024), and GPT-5 (OpenAI, 2025a) by OpenAI. For open-source LLMs, we evaluate numerous models including LLaMA3.1 (AI@Meta, 2024), LLaMA3.3, Qwen2.5 (Team, 2024a;b), Qwen3 (Yang et al., 2025), GLM4 (GLM et al., 2024), DeepSeek (Liu et al., 2024).

## 3.2 RESULTS ON ASYNCTOOL

We conducted a comprehensive evaluation of the current mainstream models. Based on these findings, our analysis is structured around three key questions.

### Q1: Which Model is Better in Completing Multiple Tasks Asynchronously?

As shown in Table 10, GPT-4.1 demonstrates the strongest performance in asynchronous capability under asynchronous multitasking evaluation, achieving a score of 38.06. Close behind, the large open-source model DeepSeek-V3.1-Terminus achieves performance comparable to closed-source models, highlighting its competitive strength.

In the step-level evaluation, closed-source models consistently achieve high scores, while open-source models exhibit notable discrepancies. This highlights the differences in the asynchronous capabilities of these models. In the sub-task evaluation, the models' scores are nearly double those of the overload models.

### Q2: What Challenges do LLMs Encounter in ASYNCTOOL?

In the ASYNCTOOL evaluation, we carefully examine the models' tool-call trajectories and conduct a detailed analysis of different numbers and classes of tasks. Models with lower performance often exhibit a lack of temporal reasoning ability, executing the next function call immediately without waiting for the tool's response to the current task. This behavior is particularly problematic when there is a strong dependency between the two calls, frequently resulting in unforeseen errors. In contrast, higher-performing models can interleave the execution of different tasks, enabling them to advance multiple tasks in parallel successfully and thereby creating a substantial performance gap.

We also observe that agents occasionally fail to complete certain tasks. Specifically, some models tend to execute the most recently presented task first, neglecting earlier tasks. This issue is more common in smaller models and is almost absent in larger 70B-scale models. Moreover, it occurs more frequently in tri-task combinations than in dual-task combinations, which is consistent with intuition. Another notable error is tool misidentification, in which models misjudge the tool use across tasks, for example, invoking a flight-booking tool when they should continue a data-processing task. This type of error typically leads to cascading failures, as the model often struggles to self-correct once the confusion occurs. Common errors, by contrast, include non-compliant instructions, erroneous tool-call trajectories, and parameter errors, all of which directly prevent tasks from being scored successfully. Such errors still constitute a significant proportion of failures in smaller models within ASYNCTOOL, but their frequency decreases substantially as model size increases.

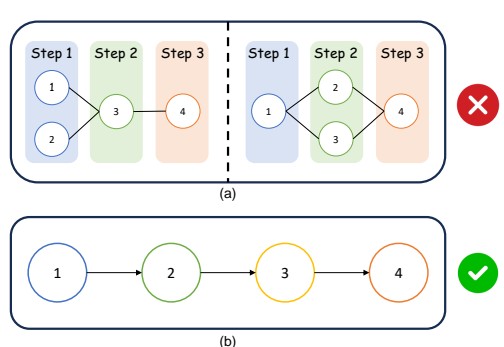

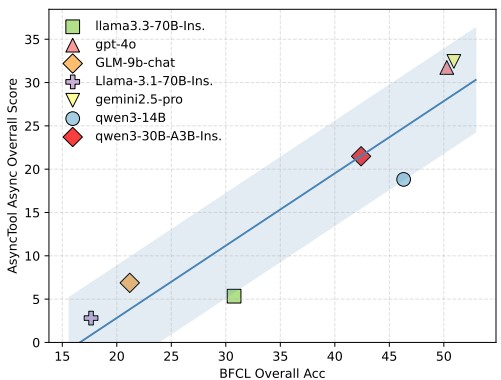

Figure 4: (a) shows the Non-deterministic trajectory. (b) shows the Deterministic trajectory.

Figure 5: Comparison between BFCL Overall Accuracy and ASYNCTOOL Overall Scores across several models.

## Q3: What Factors Make Multi-Tasking Hard in Asynchronous Tool Calls?

As shown in Figure 3, the results show that asynchronous multitasking causes varying degrees of performance degradation across different models compared with regular multitasking, accompanied by a simultaneous decline in the SC index. In terms of the SC metric, Gemini 2.5 Pro exhibits the largest drop among closed-source models, while Qwen3-8B experiences the greatest decrease among open-source models.

The fundamental difference between asynchronous and regular multitasking lies in whether the agent can obtain tool responses immediately. In our evaluation, each function call in asynchronous multitasking incurs a one-round response delay, meaning that after invoking a function, the agent cannot immediately acquire the necessary information to proceed with the subsequent step. Such information may include confirmation signals from file operations, query results, or essential state data. Generally, higher-performing models can effectively shift their attention to other tasks by invoking different tools to make progress while waiting. By contrast, lower-performing models often lack temporal awareness and inappropriately proceed to the next step of the current task. In doing so, they sometimes fabricate parameters for the current call based on imagined results from the previous call—a typical hallucination phenomenon observed in large language models.

Another factor influencing difficulty is **task quantity**. An intuitive hypothesis is that as the number of tasks increases, the overall difficulty also increases. When the number of tasks reaches four, the difficulty of each task increases significantly, and the impact of task quantity becomes more pronounced. Consequently, we do not consider a larger number of tasks in this analysis. Based on ASYNCTOOL, we find that when the number of tasks goes from two to three, the closed-source model Gemini 2.5 Pro decreases the most, from 42.56 to 32.44 while the open-source model Qwen3-8B decreases the most, from 27.91 to 10.67 while the closed-source model GPT-4.1 and the open-source model LLaMA-3.1-70B-Ins decrease less. Full results are listed in Appendix C.1

### 3.3 DISCUSSION

In comparing ASYNCTOOL with an established tool-use benchmark, we observe a broadly similar trend in model performance as shown in Figure 5. Models that perform better in traditional tool-use settings also tend to achieve higher scores in asynchronous multi-task scenarios. This correlation may be explained by the fact that both types of tasks require similar underlying competencies, particularly the ability to determine intra-task step dependencies and retain parameter information across interactions, alongside robust context management and coherent integration of outputs.

Unlike conventional single-task, immediate-response tool-use benchmarks, ASYNCTOOL evaluates agents from a complementary perspective. It not only measures how well a model can invoke tools, but also assesses its capacity to maintain intermediate parameters in memory while resolving intra-task step dependencies and managing inter-task parallelism. This design imposes additional

reasoning demands compared with traditional evaluations, as the model must track and coordinate multiple, partially interdependent processes under realistic latency constraints.

These differences make ASYNCTOOL a valuable supplement to existing benchmarks. It provides insight into dimensions of tool-augmented reasoning that are often overlooked in standard settings, such as memory retention under long-horizon tasks and the ability to plan actions based on dependency structures. We hypothesize that architectural or training strategies aimed at improving asynchronous multitasking, for example, by enhancing execution planning mechanisms or introducing latency-aware scheduling policies, might not only enhance performance in ASYNCTOOL but could also yield measurable improvements in conventional tool-use benchmarks. If this conjecture holds, it would underscore the strategic importance of emphasizing asynchronous mechanisms in the design of next-generation AI agents.

## 4 RELATED WORK

**Multi-Step Tool Call.** As LLM-based agents continue to improve in their ability to leverage external tools, related benchmarks have higher complexity and richer dependencies. Some tasks now require multiple tool calls, executed in the correct order, to be completed, which presents significant challenges for the models in terms of comprehension and planning. Currently, two primary approaches are used to evaluate this capability. The first approach asks the model to provide, in a single step, the full sequence of tool calls needed to solve the task, focusing on the model's ability to plan tool usage (Yu et al., 2025b; Huang et al., 2024; Zhang et al., 2024b; Chen et al., 2024a; Wu et al., 2024). The second approach allows the model to invoke tools step by step, receiving responses at each stage. It emphasizes evaluating the multi-step tool calling process until task completion (Yan et al., 2024; Basu et al., 2024; Zhong et al., 2025).

The order of tool calls is often a critical aspect to evaluate. (Yan et al., 2024) constrains the tool call sequence using traditional rule-based restrictions, while NESTFUL (Basu et al., 2024) enforces a fixed call order through parameter dependencies. A key limitation of existing benchmarks is their failure to account for tool invocation delays frequently encountered in real-world applications. ASYNC-TOOLaddresses this gap, enabling systematic evaluation of a model's capacity for asynchronous tool calling.

**Asynchronous Multitasking Scenarios.** The ability to perform multiple tasks asynchronously is an inevitable trend in the development of large LLMs. (Chu et al., 2023; Tan et al., 2023; Wang & Zhao, 2023; Ginart et al., 2024) introduce time changes in model evaluation and provide related evaluation tasks.(Zhang et al., 2024a) introduces the concept of time in multitasking, aiming to assess the efficiency with which models manage multiple tasks concurrently. (Gonzalez-Pumariega et al., 2025) evaluates models' asynchronous planning, focusing on failure modes and challenges in integrating long-term information. While important for understanding LLMs in asynchronous tasks, these studies are limited to simulations and don't assess performance on tasks requiring real-world tool calls. Moreover, (Gim et al., 2024) enhances the operational efficiency of LLMs by enabling them to generate and execute function calls concurrently, representing a novel approach. Unlike prior benchmarks, ASYNCTOOL is the first to systematically evaluate models' capabilities for asynchronous tool calling in realistic multi-task scenarios, explicitly incorporating tool response latency into the assessment.

## 5 CONCLUSION

In this paper, we present ASYNCTOOL, the first benchmark to explicitly incorporate the notion of time into the evaluation of agent tool use. ASYNCTOOL captures not only the intra-task step dependencies within a task but also the associated tool response latency–an essential factor that drives agents to perform multiple tasks efficiently under an asynchronous framework in realistic scenarios. Leveraging this asynchronous tool calling paradigm, ASYNCTOOL systematically evaluates LLM-based agents in terms of their asynchronous multitasking capabilities and their ability to perform tool calling under realistic latency constraints. Experimental results show that as the number of tasks increases and their compositions become more complex, model performance degrades significantly, with many failing to complete the tasks successfully. These findings suggest that tool-using agents still have considerable

room for improvement in handling multiple tasks asynchronously, providing valuable guidance for the advancement of future agent systems.

## REPRODUCIBILITY STATEMENT

Our work is fully reproducible. We guarantee the reproducibility of all results and the reliability of the models used. All data and code will be released within one month after the article is accepted.

## ETHICS STATEMENT

Our paper proposes a new LLM benchmark that contains frequently updated questions from new information sources, scores answers based on objective ground truth, and covers a wide variety of tasks. We believe our work does not have any inherent negative impact on society. We hope that our work will have a positive impact on practitioners and researchers: by providing a new benchmark with frequently updated questions, our work has the potential to both accelerate future research and enable more comprehensive and rigorous evaluations of existing and future models. Furthermore, we hope that the overall framework of our benchmark—containing frequently updated questions from new information sources—will become popular, thereby mitigating the negative impact of contamination in future LLM assessments and making LLM benchmarks more "future-proof."

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

## A   EVALUATION

### A.1   METRICS

**Character matching.**We extract the function name and parameters and match them against the golden truth. We also consider the possibility that the model might execute some incorrect function calls due to the use of subset validation. Specifically, given a tool list $T$ and a query $q$, the model generates a series

$$\mathcal{T}^{\text{pred}} = \langle a_1^{\text{pred}}, a_2^{\text{pred}}, \ldots, a_n^{\text{pred}} \rangle,$$

where $a_i^{\text{pred}}$ is the predicted LLM action at turn $i$. The golden truth is

$$\mathcal{T}^{\text{gt}} = \langle a_1^{\text{gt}}, a_2^{\text{gt}}, \ldots, a_n^{\text{gt}} \rangle.$$

We verify whether $A^{\text{gt}} \subseteq A^{\text{pred}}$.

**F1 function & Parameters.**Referring to the implementation of NESTFUL(Basu et al. (2024)), we use the above metrics to evaluate the model's responses.In our experiments, we find that frequent task switching by the model interferes with its ability to recognize different tasks accurately. Specifically, the model often invokes incorrect tools for the tasks, which leads to a decline in performance metrics.

**Trajectory completion.**After we extract the function, we will throw $\mathcal{T}^{gt}$ and $\mathcal{T}^{pred}$ to the executor for execution. The executor will return the corresponding results $R^{gt}$ and $R^{pred}$, and we will compare the results using the subset judgment method the same as above.

**Environment matching.**Referring to the implementation of BFCL(Yan et al. (2024)), we use the executor class instance for comparison.In our data, some tool calls induce alterations to the environment, while the majority of tool calls do not result in any environmental changes, such as those associated with queries. Nonetheless, we incorporate this metric into the evaluation to ensure comprehensiveness.

Table 3: **Abbreviation explanation of data categories.**

| Full Name | Abbreviation |
| --- | --- |
| Data Management | DM |
| Filesystem | FS |
| Data generation | DG |
| MessageAPI | MA |
| Number operations | NO |
| SocialConnect | SC |
| String Manipulation | SM |
| TicketPurchase | TP |
| TradingBot | TB |
| TravelPlanning | TP* |
| DataFormat | DF |
| Machine Operation | MO |

### A.2   MODEL VERSION

The version for GPT-4o is `gpt-4o-2024-11-20`, for GPT-4.1 is `gpt-4.1-2025-04-14`, and for Gemini 2.5 Pro is `gemini-2.5-pro-preview-05-06`.The version for GLM-4-32B is `GLM-4-32B-0414`.The vesion for Qwen3-30B-A3B-Ins. is Qwen3-30B-A3B-Instruct-2507.

## B   DATA

### B.1   DATA DETAILS

As shown in Table 3, the content inside introduces the meaning of our categories. Besides, Table 5 shows the data distribution of ASYNCTOOL.

Table 4: **Task Counts and Average Trajectory Lengths.**

| Type | DM | FS | DG | MA | NO | SC | SM | TP | TB | TP* | DF | MO |
|------|-----|------|------|------|------|------|------|------|------|------|------|------|
| #Tasks | 59 | 53 | 29 | 23 | 24 | 18 | 34 | 21 | 15 | 20 | 44 | 18 |
| Avg. len. | 2.14 | 3.58 | 2.14 | 3.48 | 2.12 | 4.11 | 2.09 | 2.81 | 3.27 | 3.10 | 2.09 | 3.00 |

## B.2 DATA LENGTH

We counted the input and output lengths of each task to ensure that they were within a reasonable range. These data are shown in Figure 6.

## B.3 SUPPLEMENTARY DATA

For the four-task test, considering the large number of combination types and tasks, we randomly screened the data after each combination and finally selected 300 SIMILAR and 300 CROSS four-task data for comparison. The results are only used for ablation experiments.

Table 5: **Data composition of ASYNCTOOL.** # denotes number of, and % denotes proportion of. SIMILAR means similar task combinations and CROSS means cross task combinations.

| Method | #Tasks | #Samples | %Samples |
|--------|--------|----------|----------|
| SIMILAR | 2 | 120 | 16.85% |
| CROSS | 2 | 132 | 18.54% |
| SIMILAR | 3 | 240 | 33.71% |
| CROSS | 3 | 220 | 30.90% |

# C RESULTS

In this section, we present the comprehensive results, categorized by different types, along with the test outcomes for varying numbers of tasks on the open-source model.

## C.1 DETAILED RESULTS

As shown in Figure 6, we compile and present the detailed results by task type.

## C.2 OTHER RESULTS

As shown in Figure 7, we conduct a preliminary experiment on the open-source model to demonstrate the impact of the number of tasks on the score.

## C.3 ANALYSIS OF ERROR CASES

We provide numerous examples of common model errors in the end of appendix. A frequent issue, likely due to insufficient prior training on multiple tasks, is the model forgetting about ongoing tasks. This results in a high rate of incompletion failures, despite the model's high accuracy in completing subtasks.In open-source models, frequent task switching presents an even greater challenge to the model's memory and localization abilities, causing confusion in function calls, which is a primary source of errors.

## C.4 ANALYSIS OF MODEL PERFORMANCE

We must acknowledge that even the most advanced models perform suboptimally on our benchmark. However, we need to analyze the reasons behind this from two perspectives. First, we have conducted ablation experiments on mainstream large models, revealing that as the number of tasks increases, the

Table 6: **Detailed Results of ASYNCTOOL.** We approach the analysis from the perspective of dataset classification to calculate the results. We find that the scores across different categories exhibit significant disparities. **Bold** indicates best overall performance, while underline denotes the best within the same group.

| Models | SIMILAR#2 | SIMILAR#3 | CROSS#2 | CROSS#3 | Overall |
|---|---|---|---|---|---|
| *Closed-Sourced Models* | | | | | |
| Qwen-max | 32.50 | 15.42 | 41.67 | 23.18 | 25.56 |
| Kimi-K2 | 33.33 | 20.83 | 30.30 | 20.00 | 24.44 |
| Gemini 2.5 Pro | 44.17 | 27.5 | 40.15 | 26.82 | 32.44 |
| GPT-5 | 41.67 | 21.67 | 43.94 | **28.64** | 31.32 |
| GPT-4o | 42.50 | 29.17 | 41.67 | 22.73 | 31.74 |
| GPT-4.1 | **50.00** | **36.67** | **47.73** | 27.27 | **38.06** |
| *Open-Sourced Models < 20B* | | | | | |
| LLaMA-3.1-8B-Ins. | 1.67 | 0.42 | 4.55 | 0 | 1.26 |
| GLM-4-9B-chat | 12.50 | 7.08 | 8.33 | 2.73 | 6.88 |
| Qwen2.5-7B-Ins. | 10.00 | 4.17 | 12.12 | 2.27 | 6.04 |
| Qwen2.5-14B-Ins. | 30.00 | 17.92 | 18.94 | 13.64 | 18.82 |
| Qwen3-8B | 25.83 | 5.42 | 15.91 | 5.00 | 10.67 |
| Qwen3-14B | 27.50 | 14.58 | 28.03 | 13.18 | 18.82 |
| *Open-Sourced Models >20B* | | | | | |
| LLaMA-3.1-70B-Ins. | 5.83 | 1.67 | 6.06 | 0.45 | 2.81 |
| LLaMA-3.3-70B-Ins. | 10.00 | 3.75 | 8.33 | 2.73 | 5.34 |
| GLM-4-32B | 21.67 | 13.75 | 21.97 | 9.09 | 15.17 |
| Qwen3-32B | 31.67 | 13.33 | 26.52 | 14.09 | 19.10 |
| Qwen2.5-32B-Ins. | 34.17 | 20.83 | 31.82 | 20.00 | 24.86 |
| Qwen3-30B-A3B-Ins. | 30.00 | 15.42 | 32.58 | 16.82 | 21.49 |
| DeepSeek-V3.1-Terminus | 32.50 | 24.58 | 35.61 | 27.73 | 28.93 |

Table 7: **Experimental results with different numbers of tasks on SYNC setting.**
In #number,number means the number of tasks. # 3 is compared with # 2 and # 4 is compared with # 3. The data source of # 4 is shown in Appendix B.1

| Models | # 2 | | | # 3 | | | | # 4 | | | |
|---|---|---|---|---|---|---|---|---|---|---|---|
| | S#2 | C#2 | Overall | S#3 | C#3 | Overall | Drop | S#4 | C#4 | Overall | Drop |
| Qwen2.5-7B-Ins. | 20.00 | 14.39 | 17.06 | 11.67 | 10.00 | 10.87 | 36.28% | 8.51 | 3.44 | 6.33 | 41.77% |
| GLM-4-9b-chat | 20.83 | 21.97 | 21.43 | 14.58 | 10.91 | 12.83 | 40.13% | 12.73 | 3.65 | 8.17 | 36.32% |
| Qwen2.5-72B-Ins. | 53.33 | 47.73 | 50.79 | 35.42 | 31.82 | 33.70 | 33.65% | 20.33 | 23.67 | 22.00 | 34.72% |
| LLaMA3.3-70B-Ins. | 25.00 | 16.67 | 20.63 | 10.42 | 6.82 | 8.70 | 57.83% | 3.67 | 3.67 | 3.67 | 57.81% |

difficulty grows nonlinearly. To encourage future models to achieve greater capabilities, we designed the benchmark such that tasks involving three or more objectives account for over 60

## C.5 ABLATION RESULTS

# D USE OF LARGE LANGUAGE MODELS

We use a large language model for translation and language polishing, but its role in the paper is limited. The core research and content are completed by us.

Table 8: **Supplementary Results of AsyncTool.All experiments are conducted with a delay of two turns. Bold** indicates best overall performance, while underline denotes the best within the same group.

| Models | Step Level | | Sub-Task Level | | | Task Level | | Overall |
| --- | --- | --- | --- | --- | --- | --- | --- | --- |
| | Func. | Param. | Char. | Env. | Acc. | Char. | Env. | |
| *Closed-Sourced Models* | | | | | | | | |
| Qwen-Max | **95.81** | **82.70** | **64.7** | **77.18** | **60.18** | **35.11** | 53.51 | **30.76** |
| Gemini 2.5 Pro | 91.94 | 80.05 | 57.48 | 75.90 | 56.48 | 26.97 | 53.65 | 25.70 |
| GPT-4o | 91.41 | 79.79 | 57.70 | 75.42 | 56.79 | 27.67 | 53.37 | 26.54 |
| GPT-4.1 | 91.32 | 79.97 | 57.96 | 75.90 | 6.90 | 27.67 | **54.49** | 26.40 |
| *Open-Sourced Models < 20B* | | | | | | | | |
| LLaMA-3.1-8B-Ins. | 83.16 | 44.42 | 13.38 | 40.76 | 12.42 | 1.54 | 14.75 | 1.40 |
| Qwen2.5-7B-Ins. | 87.67 | 66.97 | 32.27 | 58.28 | 29.99 | 6.88 | 26.40 | 5.48 |
| Qwen2.5-14B-Ins. | 86.75 | 72.18 | 50.21 | 63.59 | 45.06 | 19.94 | 35.11 | 15.17 |
| Qwen3-8B | 59.06 | 49.4 | 24.52 | 48.99 | 23.99 | 6.6 | 21.77 | 6.6 |
| *Open-Sourced Models > 20B* | | | | | | | | |
| LLaMA-3.1-70B-Ins. | 90.73 | 45.45 | 17.04 | 40.02 | 16.14 | 1.83 | 12.36 | 1.4 |
| Qwen3-32B | 70.67 | 60.99 | 34.34 | 56.95 | 33.7 | 11.52 | 30.76 | 10.96 |
| Qwen3-30B-A3B-Ins. | 95.57 | 79.72 | 55.41 | 71.71 | 52.81 | 23.17 | 44.66 | 20.65 |
| DeepSeek-V3.1-Terminus | 81.69 | 70.52 | 43.84 | 65.02 | 42.94 | 15.73 | 36.94 | 14.61 |

Table 9: **Supplementary Results ofAsyncTool.All experiments are conducted with a delay of zero to one turn randomized. Bold** indicates best overall performance, while underline denotes the best within the same group.

| Models | Step Level | | Sub-Task Level | | | Task Level | | Overall |
| --- | --- | --- | --- | --- | --- | --- | --- | --- |
| | Func. | Param. | Char. | Env. | Acc. | Char. | Env. | |
| *Closed-Sourced Models* | | | | | | | | |
| Qwen-Max | **96.65** | **84.21** | **66.30** | 80.57 | 64.65 | **36.80** | 58.43 | 34.41 |
| Gemini 2.5Pro | 95.57 | 83.85 | 65.45 | 81.69 | 64.92 | 35.81 | 62.22 | 35.25 |
| GPT-4o | 96.00 | 84.03 | 66.08 | 81.74 | **65.61** | 35.96 | 62.08 | **35.53** |
| GPT-4.1 | 95.45 | 83.58 | 65.76 | **81.90** | 65.18 | 35.67 | **62.92** | 34.55 |
| *Open-Sourced Models < 20B* | | | | | | | | |
| LLaMA-3.1-8B-Ins. | 83.93 | 47.44 | 16.24 | 43.84 | 15.13 | 1.97 | 16.71 | 1.69 |
| Qwen2.5-7B-Ins. | 83.73 | 67.22 | 32.11 | 59.02 | 31.00 | 8.71 | 29.07 | 8.43 |
| Qwen2.5-14B-Ins. | 85.00 | 73.32 | 48.30 | 67.46 | 46.18 | 19.24 | 39.47 | 17.98 |
| Qwen3-8B | 69.38 | 60.22 | 38.80 | 61.09 | 38.32 | 14.47 | 35.11 | 14.04 |
| *Open-Sourced Models > 20B* | | | | | | | | |
| LLaMA-3.1-70B-Ins. | 90.49 | 49.28 | 20.12 | 45.59 | 19.75 | 3.51 | 18.12 | 3.37 |
| Qwen3-32B | 76.21 | 67.91 | 47.45 | 66.88 | 47.03 | 20.79 | 43.54 | 20.08 |
| Qwen3-30B-A3B-Ins. | 95.10 | 82.17 | 59.39 | 76.70 | 58.86 | 26.69 | 52.67 | 26.26 |
| DeepSeek-V3.1-Terminus | 90.04 | 78.29 | 56.32 | 75.96 | 55.73 | 26.97 | 52.39 | 26.26 |

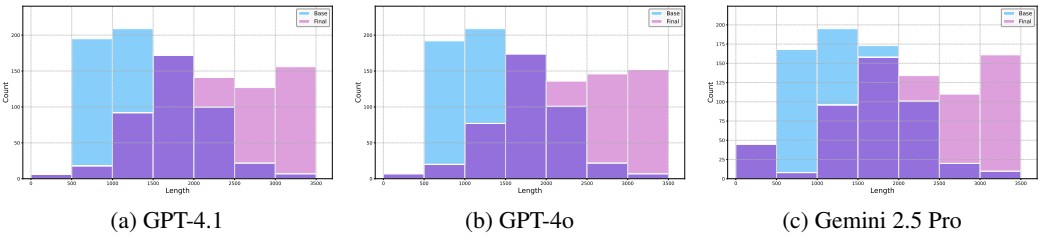

(a) GPT-4.1        (b) GPT-4o        (c) Gemini 2.5 Pro

Figure 6: Length distribution for base and final conversation, measured by the number of tokens.

Table 10: **Supplementary Results of**AsyncTool.**All experiments are conducted with a delay of one to two turns randomized. Bold** indicates best overall performance, while underline denotes the best within the same group.

| Models | Step Level | | Sub-Task Level | | | Task Level | | Overall |
|---|---|---|---|---|---|---|---|---|
| | Func. | Param. | Char. | Env. | Acc. | Char. | Env. | |
| *Closed-Sourced Models* | | | | | | | | |
| Qwen-Max | **96.87** | **83.56** | **64.49** | **78.50** | **61.94** | **33.85** | 54.63 | **30.20** |
| Gemini 2.5 Pro | 93.98 | 81.68 | 61.84 | 78.34 | 61.04 | 30.62 | **57.58** | 29.35 |
| GPT-4o | 93.08 | 81.05 | 60.51 | 78.29 | 59.66 | 30.06 | 57.44 | 28.79 |
| GPT-4.1 | 91.97 | 80.29 | 58.76 | 76.17 | 58.07 | 30.34 | 55.90 | 29.49 |
| *Open-Sourced Models < 20B* | | | | | | | | |
| LLaMA-3.1-8B-Ins. | 83.77 | 45.06 | 12.74 | 41.14 | 11.68 | 1.54 | 15.03 | 1.54 |
| Qwen2.5-7B-Ins. | 86.02 | 66.81 | 29.09 | 56.48 | 27.55 | 6.88 | 26.26 | 6.04 |
| Qwen2.5-14B-Ins. | 87.22 | 73.30 | 49.10 | 65.82 | 45.70 | 19.52 | 36.94 | 16.29 |
| Qwen3-8B | 60.72 | 50.76 | 25.85 | 51.38 | 25.42 | 7.30 | 25.14 | 7.16 |
| *Open-Sourced Models > 20B* | | | | | | | | |
| LLaMA-3.1-70B-Ins. | 90.76 | 46.37 | 16.67 | 40.98 | 16.03 | 2.25 | 13.90 | 1.83 |
| Qwen3-32B | 71.41 | 61.92 | 37.10 | 59.55 | 36.20 | 12.08 | 33.71 | 11.24 |
| Qwen3-30B-A3B-Ins. | 95.39 | 79.86 | 54.09 | 72.13 | 52.87 | 22.75 | 46.49 | 20.93 |
| DeepSeek-V3.1-Terminus | 84.52 | 73.42 | 47.35 | 68.31 | 46.07 | 17.98 | 39.89 | 16.57 |

Table 11: **Results of AsyncTool on fewshot settings.Specifically, we provided a successful trajectory in the prompt as a reference.**

| Models | Step Level | | Sub-Task Level | | | Task Level | | Overall |
|---|---|---|---|---|---|---|---|---|
| | Func. | Param. | Char. | Env. | Acc. | Char. | Env. | |
| *Open-Sourced Models < 20B* | | | | | | | | |
| Qwen2.5-7B-Ins. | 82.40 | 65.01 | 28.66 | 55.15 | 26.38 | 6.74 | 25.84 | **6.04** |
| +fewshot | 78.89 | 61.72 | 30.04 | 54.41 | 28.13 | 8.99 | 26.40 | **8.29** |
| LLaMA-3.1-8B-Ins. | 78.29 | 43.69 | 12.95 | 41.08 | 12.47 | 1.26 | 14.61 | **1.26** |
| +fewshot | 90.27 | 63.35 | 29.67 | 56.37 | 28.03 | 7.16 | 27.67 | **6.74** |
| Qwen3-8B | 63.05 | 53.61 | 29.88 | 53.08 | 29.30 | 11.10 | 28.65 | **10.67** |
| +fewshot | 72.04 | 60.67 | 34.82 | 34.82 | 34.34 | 11.52 | 32.30 | **11.24** |
| Qwen2.5-14B-Ins. | 81.32 | 70.22 | 50.21 | 65.29 | 46.28 | 22.61 | 38.20 | **18.32** |
| +fewshot | 94.28 | 80.40 | 54.19 | 75.32 | 52.87 | 22.89 | 50.14 | **21.91** |
| Qwen2.5-72B-Ins. | 95.69 | 82.78 | 65.07 | 79.78 | 62.00 | 34.41 | 58.01 | **31.04** |
| +fewshot | 89.34 | 83.61 | 66.40 | 81.16 | 64.65 | 36.94 | 59.69 | **34.55** |

Table 12: **Supplementary Results of ASYNCTOOL. Specifically, we analyzed the average number of rounds of interaction in successfully completed task models.**

| Models | # 2 | # 3 |
|---|---|---|
| *Closed-Sourced Models* | | |
| Qwen-max | 6.91 | 9.38 |
| Gemini 2.5 Pro | 6.85 | 10.50 |
| GPT-4o | 6.95 | 10.52 |
| GPT-4.1 | 6.70 | 10.57 |
| *Open-Sourced Models < 20B* | | |
| LLaMA-3.1-8B-Ins. | 9.75 | 9.00 |
| Qwen2.5-7B-Ins. | 8.50 | 12.35 |
| Qwen2.5-14B-Ins. | 7.94 | 10.95 |
| Qwen3-8B | 6.64 | 9.19 |
| *Open-Sourced Models >20B* | | |
| LLaMA-3.1-70B-Ins. | 8.41 | 16.28 |
| LLaMA-3.3-70B-Ins. | 10.62 | 16.23 |
| Qwen2.5-32B-Ins. | 9.20 | 13.68 |

**Standardization of Test Data**

**System Prompt:**
You are an expert in handling multiple tasks using functions. Your goal is to call one function in each round, gradually completing all tasks. When a function call is not yet complete, you can switch to another task. If you decide to invoke a function for a task, you MUST format it as follows:

```
{"id": "id of task", "func_name": "func_name", "params": {"params_name1":
"params_value1", "params_name2": "params_value2"}}
```

You must invoke only one function in your response. If you believe all tasks have been completed, please use the following format to reply:

```
{"content": "ALL COMPLETED"}
```

Here is a list of functions in JSON format that you can invoke.

```
[API List]
```

- - - - - - - - - - - - - - - - - - - - - - - - - - - - - - - - - - - - - - - - - - - - - - - - - - - -

**User:** Below are all tasks you need to finish:

```
{
    id:'file_13',
    question:'Compare 'academic_venture/goals.txt' with
    'academic_venture/reference_goals.txt' to verify if they are identical, then
    rename 'goals.txt' to 'master_goals.txt' in the same directory
    tory.'
}
{
    id:'SM_11',
    question have a list of integers [1, 2, 3, 4, 5]. I want to convert this list
    into a string representing a space-separated sequence of those integers. Then, I
    want to normalize this string by replacing all non-alphanumeric characters with
    underscores and converting to lowercase.'
}
{
    id:'posting_11',
    question:'Now that I'm following 'AI_enthusiast', I'd like to know more about
    them
    . Please retrieve their user statistics. After that, fetch all of their tweets.
    I remember seeing one about 'ethics'; find that specific tweet and add the
    comment 'A very important point to consider.' to it.'
}
```

**Ground True:**

```
[{
    "task_id": "file_13",
    "ground_truth": [
        "cd(folder='academic_venture')",
        "diff(file_name1='goals.txt', file_name2='reference_goals.txt')",
        "mv(source='goals.txt', destination='master_goals.txt')"
        ],
    },
    {
    "task_id": "SM_11",
    "ground_truth": [
        "integer_list_to_string(integer_list=[1, 2, 3, 4, 5])",
        "normalize_string(s='1 2 3 4 5')"
        ],
    },
    {
    "task_id": "posting_11",
    "ground_truth": [
        "get_user_stats(username='AI_enthusiast')",
        "get_user_tweets(username='AI_enthusiast')",
        "comment(tweet_id=18, comment_content='A very important point to
        consider.')"
        ],
    },
}]
```

Figure 7: An example of standardized test data.

---

**Prompt for Reconstruction**

**Prompt:**
You are an AI assistant with API-calling capabilities. Your task is to refine query descriptions based on given contexts and initial configurations, then provide explicit solution paths. You must meet these requirements:

1. Ensure problems are solvable within the given initial configuration, avoiding ambiguous or unsolvable cases. Clarify any ambiguities in the original problem by deriving explicit specifications from the initial configuration. Including but not limited to: file names, folder paths, numerical counts, and similar parameters.

2. Define solution paths as strict API call sequences (not less than 2 steps) with strong dependencies—later steps must require outputs from earlier ones. Expand the problem if necessary to meet this length requirement.

3. Preserve existing explicit information (*e.g.*, filenames, data). Extend problems only by deriving new details from the initial config.

4. Always provide the most efficient path—no unnecessary API calls. Consecutive duplicate function calls are prohibited in the execution sequence.

5. Never explicitly name APIs in the problem description. Carefully review the function description. You must ensure all function calls in the provided ground truth strictly comply with the specifications. For example, certain filesystem operations are restricted to the current working directory and cannot process multi-level paths. Only functions explicitly defined in the function description may appear in the ground truth. 6. For every input data given, you must return the same number of corresponding processed data.

6. You have permission to edit the question and ground_truth fields, and your response must explicitly return modified versions. The output format should follow the usage examples in **JSON format** instead of the origin data.

You can use the following APIs:
```
[API Document]
```
Here is one usage examples that you must follow:
```
[
    {
        "id": "number_0",
        "question":"Hey there, I need to set up a directory titled 'Reports' in my
        current workspace. Once that's in place, could you help me locate a file
        called 'summary.doc' in this vicinity and transfer it to the new 'Reports'
        folder? After that, I'd appreciate if you could find and open up 'data.txt',
        then scan through it to identify lines that mention 'Q4 financials'. Also,
        could you let me know the total number of lines in 'data.txt'?",
        "ground_truth":["mkdir(dir_name='Reports')","mv(source='summary.doc',
        destination='Reports')", "cat(file_name='data.txt')", "grep(pattern='Q4
        financials', file_name='data.txt')", "wc(file_name='data.txt',mode='l')"],
        "thought":"your tought"
    },
    {
        "id": "number_1",
        "question":"Let's bring some order to the project documents. I want to human
        readable log the storage usage of the entire current directory to usage.txt
        file. The content of the file should be the number follwed by the word bytes
        and nothing else.",
        "ground_truth":["du(human_readable=True)", "touch(file_name='usage.txt')",
        "echo(content='205 bytes', file_name='usage.txt')"],
        "thought":"your tought"
    }
]
```
where 'id' is the id of task, 'question' is the user query, 'ground_truth' is the shortest function calls sequences used to solve the query, and 'thought' is your ideas for refining this query.

Figure 8: prompt for reconstruction.

---

**Reconstructed Data**

**Task:**
Locate a file with "ideas" in its name within the 'documents' folder, display its contents, and create a backup copy named 'ideas_backup.txt' in the same directory.
**Config:**

```
{
  "root": {
    "simona": {
      "type": "directory",
      "contents": {
        "documents": {
          "type": "directory",
          "contents": {
            "ideas.txt": {
              "type": "file",
              "content": "Collaboration leads to success. Innovation ignites growth."
            },
            "Archived": {
              "type": "directory",
              "contents": {}
            },
            "past_projects": {
              "type": "directory",
              "contents": {}
            }
          }
        }
      }
    }
  }
}
```

**Trajectory:**

```
[
  "cd(folder='documents')",
  "find(name='ideas')",
  "cat(file_name='ideas.txt')",
  "cp(source='ideas.txt', destination='ideas_backup.txt')"
]
```

Figure 9: An example of reconstructed data: GorillaFileSystem.

**API Document**

```
{
    "name": "mv",
    "description": "This tool belongs to the Gorilla file system. It is a simple
    file system that allows users to perform basic file operations such as
    navigating directories, creating files and directories, reading and writing to
    files, etc. Tool description: Move a file or directory from one location to
    another. so",
    "parameters": {
        "type": "dict",
        "properties": {
            "source": {
                "type": "string",
                "description": "Source name of the file or directory to move. Source
                must be local to the current directory."
            },
            "destination": {
                "type": "string",
                "description": "The destination name to move the file or directory
                to. Destination must be local to the current directory and cannot be
                a path. If destination is not an existing directory like when
                renaming something, destination is the new file name. "
            }
        },
        "required": [
            "source",
            "destination"
        ]
    },
    "response": {
        "type": "dict",
        "properties": {
            "result": {
                "type": "string",
                "description": "The result of the move operation."
            }
        }
    }
}
```

Figure 10: An example of a API document: TravelAPI.

---

**Correct Trajectory**

**User:**
Below is all tasks you need to finish:

```
{id:'trading_0',question:'Can you get me the ticker for 'Alpha Tech', see its
current price, and then execute a purchase of 20 shares at a price of $1320.5?'}
{id:'file_11',question:'Hey, can you set up a new directory named 'Projects' right
in workspace folder?'}
```

- - - - - - - - - - - - - - - - - - - - - - - - - - - - - - - - - - - - - - - - - - - - -

**Assistant:**

```
{
    "id": "trading_0",
    "func_name": "get_symbol_by_name",
    "params": {"name": "Alpha Tech"}
}
```

**Environment:**

```
{"role": "tool", "content": "{"wait": "The current function is being executed."}"}
```

**Assistant:**

```
{
    "id": "file_11",
    "func_name": "cd",
    "params": {"folder": "workspace"}
}
```

**Environment:**

```
{"role": "tool", "content": "{"wait": "The current function is being executed."}"},
{"role": "tool", "id": "trading_0", "function name": "get_symbol_by_name(name="Alpha
Tech")", "response": "{"symbol": "ALPH"}"}
```

**Assistant:**

```
{
    "id": "trading_0",
    "func_name": "get_stock_info",
    "params": {"symbol": "ALPH"}
}
```

**Environment:**

```
{"role": "tool", "content": "{"wait": "The current function is being executed."}"},
{"role": "tool", "id": "file_11", "function name": "cd(folder="workspace")",
"response": "{"current_working_directory": "workspace"}"}
```

**Assistant:**

```
{
    "id": "file_11",
    "func_name": "mkdir",
    "params": {"dir_name": "Projects"}
}
```

**Environment:**

```
{"role": "tool", "content": "{"wait": "The current function is being executed."}"},
{"role": "tool", "id": "trading_0", "function name":
"get_stock_info(symbol="ALPH")", "response": "{"price": 1320.45, "percent_change":
-0.08, "volume": 1.567, "MA(5)": 1321.12, "MA(20)": 1325.78}"}
```

**Assistant:**

```
{
    "id": "trading_0",
    "func_name": "place_order",
    "params": {"order_type": "Buy", "symbol": "ALPH", "price": 1320.5, "amount": 20}
}
```

**Environment:**

```
{"role": "tool", "content": "{"wait": "The current function is being executed."}"},
{"role": "tool", "id": "file_11", "function name": "mkdir(dir_name="Projects")",
"response": "None"}
```

**Assistant:**

```
{"content": "ALL COMPLETED"}
```

Figure 11: An example of correct trajectory.

```
Error Caused by Insufficient Temporal Awareness

User:
Below is all tasks you need to finish:
    {
        id:'trading_0',
        question:'Can you get me the ticker for 'Alpha Tech', see its current price, and
        then execute a purchase of 20 shares at a price of $1320.5?'
    }
    {
        id:'file_11',
        question:'Hey, can you set up a new directory named 'Projects' right in
        workspace folder?'
    }
- - - - - - - - - - - - - - - - - - - - - - - - - - - - - - - - - - - - - - - - - - - - - -
Assistant:
    {
        "id": "trading_0", "func_name": "get_symbol_by_name",
        "params": {"name": "Alpha Tech"}
    }
Environment:
    {
        "role": "tool",
        "content": "{"wait": "The current function is being executed."}"
    }
Assistant:
    {
        "id": "trading_0", "func_name": "get_stock_info",
        "params": {"symbol": "ATGL"}
    }
```

Figure 12: An example of an error caused by insufficient temporal awareness. In this case, the agent prematurely assumed the symbol of Alpha Tech to be "ATGL" before receiving the actual call result.

```
Error Caused by Tool Confusion

User:
Below is all tasks you need to finish:
    {
        id:'trading_0',
        question:'Can you get me the ticker for 'Alpha Tech', see its current price, and
        then execute a purchase of 20 shares at a price of $1320.5?'
    }
    {
        id:'file_11',
        question:'Hey, can you set up a new directory named 'Projects' right in
        workspace folder?'
    }
- - - - - - - - - - - - - - - - - - - - - - - - - - - - - - - - - - - - - - - - - - - - - -
Assistant:
    {
        "id": "trading_0", "func_name": "get_symbol_by_name",
        "params": {"name": "Alpha Tech"}
    }
Environment:
    {
        "role": "tool",
        "content": "{"wait": "The current function is being executed."}"
    }
Assistant:
    {
        "id": "trading_0", "func_name": "cd",
        "params": {"folder": "workspace"}
    }
```

Figure 13: An example of an error caused by tool confusion. In this case, the agent mistakenly applied the tool intended for task "file_11" to task "trading_0".

---

**Error Caused by Neglecting a Certain Task**

**User:**
Below is all tasks you need to finish:

```
{
    id:'file_13',
    question:'Compare 'academic_venture/goals.txt' with
    'academic_venture/reference_goals.txt' to verify if they are identical, then
    rename 'goals.txt' to 'master_goals.txt' in the same directory.'
}
{
    id:'SM_11',
    question:'I have a list of integers [1, 2, 3, 4, 5]. I want to convert this list
    into a string representing a space-separated sequence of those integers. Then, I
    want to normalize this string by replacing all non-alphanumeric characters with
    underscores and converting to lowercase.'
}
{
    id:'posting_11',
    question:'Now that I'm following 'AI_enthusiast', I'd like to know more about
    them. Please retrieve their user statistics. After that, fetch all of their
    tweets. I remember seeing one about 'ethics'; find that specific tweet and add
    the comment 'A very important point to consider.' to it.'
}
```

- - - - - - - - - - - - - - - - - - - - - - - - - - - - - - - - - - - - - - - - - - - -

**Assistant:**
```json
{"id": "file_13", "func_name": "cd", "params": {"folder": "academic_venture"}}
```
**Environment:**
```json
{"role": "tool", "content": "{"wait": "The current function is being executed."}"}
```
**Assistant:**
```json
{"id": "SM_11", "func_name": "integer_list_to_string", "params": {"integer_list":
[1, 2, 3, 4, 5]}}
```
**Environment:**
```json
{"role": "tool", "content": "{"wait": "The current function is being
executed."}"},{"role": "tool", "id": "file_13", "function name":
"cd(folder="academic_venture")", "response": "{"current_working_directory":
"academic_venture"}"}
```
**Assistant:**
```json
{"id": "file_13", "func_name": "diff", "params": {"file_name1": "goals.txt",
"file_name2": "reference_goals.txt"}}
```
**Environment:**
```json
{"role": "tool", "content": "{"wait": "The current function is being
executed."}"},{"role": "tool", "id": "SM_11", "function name":
"integer_list_to_string(integer_list=[1, 2, 3, 4, 5])", "response": "1 2 3 4 5"}
```
**Assistant:**
```json
{"id": "SM_11", "func_name": "normalize_string", "params": {"s": "1 2 3 4 5"}}
```
**Environment:**
```json
{"role": "tool", "content": "{"wait": "The current function is being
executed."}"},{"role": "tool", "id": "file_13", "function name":
"diff(file_name1="goals.txt", file_name2="reference_goals.txt")", "response":
"{"diff_lines": "- Research topic selection\\\\n+ Data analysis\\\\n- Literature
review\\\\n+ Data collection\\\\n- Data collection\\\\n+ Draft writing\\\\n- Data
analysis\\\\n+ Final submission\\\\n- Draft writing\\\\n+ Literature review\\\\n-
Final submission\\\\n+ Research topic selection"}"}
```
**Assistant:**
```json
{"id": "file_13", "func_name": "mv", "params": {"source": "goals.txt",
"destination": "master_goals.txt"}}
```
**Environment:**
```json
{"role": "tool", "content": "{"wait": "The current function is being
executed."}"},{"role": "tool", "id": "SM_11", "function name":
"normalize_string(s="1 2 3 4 5")", "response": "1_2_3_4_5"}
```
**Assistant:**
```json
{"content": "ALL COMPLETED"}
```

Figure 14: An example of error caused by neglecting a certain task. In this case, the task "file_13" was ignored by the agent

