# OpenReview forum: "AsyncTool: Evaluating the Asynchronous Function Calling Capability under Multi-Task Scenarios"
_ICLR.cc/2026/Conference — Submitted to ICLR 2026_

### Official Review · Reviewer_adPW · 2025-10-25

**Soundness:** 3
**Presentation:** 2
**Contribution:** 3
**Rating:** 4
**Confidence:** 3

**Summary:**

ASYNCTOOL introduces a benchmark for evaluating LLM’ ability to handle asynchronous, multi-task tool use, where tool responses are delayed. The authors construct the dataset by combining 2–3 independent tool-based tasks and simulating one-round response latencies, creating ground-truth trajectories that utilize idle time. Evaluation on top models shows that they struggle with this temporal coordination, exposing a gap in current agentic reasoning.

**Strengths:**

The paper addresses a timely and underexplored gap in agentic LLMs.

The motivation and novelty are clear: testing agents’ temporal reasoning and their ability to manage idle time.

The authors conduct a solid set of experiments across diverse models and uncover systematic failure modes.

**Weaknesses:**

1- The benchmark assumes a uniform one-turn delay for all tool calls. In practice, real-world tools have variable latencies. The paper lacks ablations or robustness analysis under different delay distributions, which limits generalizability.

2- Although ASYNCTOOL introduces three levels of evaluation (step, subtask, task-level), these are still based on correctness metrics. There is no explicit metric to quantify how well an agent utilizes idle time or performs efficient scheduling. As a result, true gains in scheduling behavior are not isolated.

3- Presentation issues:
Table 2 is difficult to interpret; column labels are not clearly defined. Also, the paper does not clearly explain how tool delays are simulated during evaluation (distinct from dataset construction). Qualitative examples showing how delayed or repetitive tool calls from the agent lead to the observed drop in metrics would help readers better connect the evaluation setup to the failure patterns.

4- The benchmark assumes a single reference trajectory that represents the optimal schedule under fixed latency. However, there can be multiple valid ways to use idle time effectively. The evaluation may penalize models that produce alternative but equally efficient schedules, making the metric sensitive to arbitrary ordering choices rather than genuine reasoning quality.

5- The paper does not explore potential solutions. Exploring mitigation strategies, like a basic fine-tuning experiment, would strengthen the contribution.

**Questions:**

See in the weaknesses above.

---

> ### Author Response · Authors · 2025-11-21
> **Response to Reviewer adPW [1/5]**
>
> We sincerely appreciate your constructive comments and insightful suggestions. Below, we respond to each concern in detail and provide further analysis and clarification.
> > **Q1**: The benchmark assumes a uniform one-turn delay for all tool calls. In practice, real-world tools have variable latencies. The paper lacks ablations or robustness analysis under different delay distributions, which limits generalizability.
>
> ### **A1**:
> We thank the reviewers for their valuable feedback on the uniform delay assumption in the benchmarking. We fully understand that delays in practical applications are often complex and diverse, and this is indeed a limitation of current research.
>
> **Delay Distribution and Robustness Analysis:** To better observe the impact of delay, we present experimental results under different delay settings in **Tables 8, 9, and 10** of the appendix, including configurations with two delays, zero to one random delay, and one to two random delays. By comparing these results, we note that random delays lead to significant fluctuations in results. This randomness introduces considerable variability into the benchmark and may also affect the fairness of the evaluation. Therefore, while these results provide valuable information as additional experiments, we have not used them as the final benchmark.
>
> **Delay Simplification and Evaluation Feasibility:** We fully acknowledge that the distribution of delays in reality is complex and variable. However, the main goal of this paper is to simplify this complexity to focus on testing the model's performance under relatively uniform delay conditions. This simplification provides us with a controlled experimental environment, facilitating more systematic error analysis and result comparison. We chose a single-round delay design to reduce uncertainty in the experiment while ensuring the reproducibility and comparability of the evaluation results.
>
> **Future Outlook:** Although our current baseline delay setting is relatively simplified, our initial intention was to provide a new perspective for future, more complex, and realistic baselines. We believe that as research progresses, considering more complex delay distributions will become a necessary direction, and we hope this research can provide some inspiration for exploring this area.

---

> ### Author Response · Authors · 2025-11-21
> **Response to Reviewer adPW [2/5]**
>
> > **Q2**: Although ASYNCTOOL introduces three levels of evaluation (step, subtask, task-level), these are still based on correctness metrics. There is no explicit metric to quantify how well an agent utilizes idle time or performs efficient scheduling. As a result, true gains in scheduling behavior are not isolated.
>
> ### **A2**:
> We thank the reviewer for this crucial comment. In ASYNCTOOL, the model needs to handle multiple tasks simultaneously, and due to the fixed latency of tool calls, the model cannot advance the same task continuously. Therefore, the model must recognize this limitation and proactively switch to other tasks, thus reflecting its time awareness and utilization of idle time to some extent.
>
> To partially alleviate this deficiency, we have added the average number of interaction rounds the model completes tasks in **Table 12**. This metric can reflect the model's task planning ability to some extent: **stronger models can usually complete tasks in fewer rounds under the same conditions**, which indirectly reflects more reasonable scheduling behavior. However, we also recognize that this metric still cannot completely separate scheduling efficiency from overall inference ability.We are very grateful to the reviewer for pointing out the importance of this direction. In future work, we plan to further explore how to construct dedicated scheduling efficiency metrics, such as explicitly measuring the model's utilization of waiting time, the optimality of scheduling strategies, or the waste of idle time, to more systematically characterize the model's planning ability in asynchronous environments.

---

> ### Author Response · Authors · 2025-11-21
> **Response to Reviewer adPW [3/5]**
>
> > **Q3**: Presentation issues: **Table 2** is difficult to interpret; column labels are not clearly defined. Also, the paper does not clearly explain how tool delays are simulated during evaluation (distinct from dataset construction). Qualitative examples showing how delayed or repetitive tool calls from the agent lead to the observed drop in metrics would help readers better connect the evaluation setup to the failure patterns.
>
> ### **A3**:
> First, regarding the column labels and their meanings in **Table 2**, we provide a preliminary explanation in **Section 2.3** of the main text and supplement it with a complete and formal label definition in **Appendix A.1** to enable readers to more accurately understand the relevant metrics. Furthermore, we have further optimized the table descriptions in the latest version to improve readability.
>
> Second, regarding the simulation method of latency, we emphasize in the paper that the injection of tool latency occurs during the evaluation phase, not the data construction phase. Latency is dynamically implemented by delaying tool returns during model inference.
>
> In addition, we fully agree with the reviewers' suggestion to provide more qualitative examples to demonstrate how latency affects model behavior. In **Figures 12, 13, and 14** of the appendix, we provide several typical failure modes, including tool invocation errors, task ignoring, and planning failure. These examples aim to help readers more intuitively understand the relationship between latency settings and performance degradation.

---

> ### Author Response · Authors · 2025-11-21
> **Response to Reviewer adPW [4/5]**
>
> > **Q4**: The benchmark assumes a single reference trajectory that represents the optimal schedule under fixed latency. However, there can be multiple valid ways to use idle time effectively. The evaluation may penalize models that produce alternative but equally efficient schedules, making the metric sensitive to arbitrary ordering choices rather than genuine reasoning quality.
>
> ### **A4**:
> First, deterministic trajectories serve only as a reference for single tasks. In multi-task scenarios, the optimal trajectory is not unique. Each multi-task dataset typically has multiple feasible and equally efficient execution paths. The model can achieve a score as long as it can reasonably plan the task sequence in an asynchronous environment, make consistent decisions based on tool returns and task states, and complete each task sequentially.
>
> Second, our evaluation does not rely on the correctness of a single path. In **Appendix A.1**, we explicitly provide multiple verification methods, including verification of the environment execution configuration and verification of the environment state after the model output is executed. These mechanisms ensure that as long as the trajectory generated by the model semantically completes all tasks, its performance can be correctly identified and will not be misjudged due to differences from the reference trajectory. Here are two examples
> ```
> step1: get_main_diagonal(matrix=[[1, 2, 3], [4, 5, 6], [7, 8, 9]])
> step2: trim_trailing_dot(string="Hello, world.")
> step3: unary_operator(x=[1, 5, 9])
> step4: find_start_end_indices(input_str="Hello, world", sub_str="world")
> step5: ALL COMPLETED
> ```
> and
> ```
> step1: trim_trailing_dot(string="Hello, world.")
> step2: get_main_diagonal(matrix=[[1, 2, 3], [4, 5, 6], [7, 8, 9]])
> step3: find_start_end_indices(input_str="Hello, world", sub_str="world")
> step4: `unary_operator(x=[1, 5, 9])`
> step5: ALL COMPLETED
> ```
> Their final results are consistent across various validation methods, including trajectory evaluation, environment evaluation, and setup checks.
>
> Furthermore, while multiple scheduling schemes may be reasonable, the model still needs to make effective planning decisions in an asynchronous, delayed environment. Therefore, we added the average number of interaction rounds as a supplementary metric in **Table 12** to reflect the efficiency of the scheduling strategy. **A stronger model reduces unnecessary rounds of calls under the same task conditions**, enabling it to complete the task in fewer rounds, providing a perspective on scheduling quality independent of the reference trajectory.
>
> In summary, our design avoids over-reliance on a single scheduling path while ensuring that different reasonable scheduling strategies can be correctly evaluated. We are very grateful to the reviewers for pointing out this important dimension, which helps us further refine the expression of the benchmark and future development directions.

---

> ### Author Response · Authors · 2025-11-21
> **Response to Reviewer adPW [5/5]**
>
> > **Q5**: The paper does not explore potential solutions. Exploring mitigation strategies, like a basic fine-tuning experiment, would strengthen the contribution.
>
> ### **A5**:
> We appreciate the valuable suggestions from the reviewers, and we highly value the exploration of potential solutions.
>
> To explore possible mitigation strategies, we present a comparison of few-shot results for some models in **Table 11**. The comparison shows that few-shot learning can improve model performance to some extent, indicating that this method has potential improvement effects in certain scenarios. Therefore, we believe it is worthwhile to further explore it as a potential solution.
>
> Furthermore, as suggested by the reviewers, we plan to conduct in-depth research and evaluation of more mitigation strategies in future work, such as fine-tuning experiments and other optimization methods. These efforts will help further improve the performance of the models and the practicality of the benchmarks.
>
> We thank the reviewers again for their comments, and we will continue to explore and incorporate these potential improvement measures into subsequent research.

---

> ### Author Response · Authors · 2025-11-27
> **Looking forward to further discussion**
>
> Dear Reviewer adPW:
>
> Thank you very much for your suggestions on our paper. We are delighted to receive your positive feedback! Thank you!
>
> We have further improved the paper based on your constructive suggestions. Below is a summary of our revisions and analysis in our response:
>
> - We clarified the evaluation objectives of the AsyncTool benchmark and added additional metrics to evaluate the model's utilization of time efficiency.
>
> - We provided a motivational analysis of the delay and more detailed experimental analyses under different experimental settings, along with a comparison of experiments under synchronous conditions.
>
> - We provided solutions to improve model performance on the benchmark and presented the experimental results.
>
> - We updated the figures and improved the paper's wording, and added relevant examples.
>
> We greatly appreciate your valuable time and expertise in reviewing our work. If possible, we would like to ask if it is possible to improve the overall score of this paper given these updates? Thank you again for your valuable insights and review!
>
> Thank you very much for your constructive comments, time, and patience.
>
> Sincerely,
>
> Authors

---

### Official Review · Reviewer_fuiW · 2025-10-31

**Soundness:** 3
**Presentation:** 3
**Contribution:** 3
**Rating:** 6
**Confidence:** 4

**Summary:**

This paper addresses a key gap in LLM agent evaluation: most benchmarks ignore tool response latency and focus only on single-task scenarios. The authors propose ASYNCTOOL, which is the first benchmark to evaluate an agent's asynchronous multitasking capabilities by simulating realistic tool response delays and requiring concurrent task execution. A high-quality dataset was also constructed using a four-step pipeline (collection, AI reconstruction, human annotation, multi-task composition). The experiments are conducted at the Step, Sub-Task, and Task levels, and show that even SOTA models struggle significantly with these complex asynchronous workflows, revealing key failure modes.

**Strengths:**

1. The authors introduce tool response latency into LLM evaluation, which is critical for real-world deployment but often overlooked in existing benchmarks.

2. The proposed three-level evaluation system (step, sub-task, task) provides a nuanced understanding of model behavior. Including 19 models across different scales strengthens its empirical contribution.

3. The pipeline for building the benchmark is transparent and well-documented, which enhances reproducibility and trustworthiness.

4. The analysis of failure modes (e.g., tool confusion, task neglect) is clear and provides specific guidance for future research.

**Weaknesses:**

1. The use of a fixed "one-round delay" is unrealistic. Real-world latency is variable and unpredictable, so the benchmark may only test a simple heuristic rather than true asynchronous management.

2. The reliance on single, deterministic ground-truth paths turns the task into "plan following" rather than "plan generation". In addition, the benchmark does not take into account resource contention or mutual exclusion (e.g., a tool being locked by one task). All of these shortages limits the benchmark's realism.

2. The current evaluations only focus on correctness (e.g., accuracy, F1), not considering efficiency (such as total task completion time, token usage, or number of tool calls), which is important in real-world multitasking scenarios.

4. The evaluations do not penalize excessive or unnecessary tool calls, which could lead to gaming the system through brute-force switching strategies.

**Questions:**

1. Could you clarify how the latency of each tool call is determined? Is it uniformly set to one round for all tools? If so, what is the justification for this simplification, and do you plan to support heterogeneous or stochastic latency in future versions?

2. Have you conducted the experiments without tool-call latency (i.e., synchronous setting)? Such a comparison would help isolate the impact of asynchrony and better demonstrate the value of the benchmark.

3. If a model is strong at single-task tool use, it might achieve high scores by frequently switching tasks with a carefully crafted prompt. Have you observed such behavior? Are there any mechanisms or penalties in place to ensure that the evaluation rewards strategic scheduling rather than heuristic over-invocation?

4. Can you elaborate on the trade-off between evaluation simplicity (using deterministic paths) and realism? Is ASYNCTOOL evaluating planning or just the ability to follow a complex instruction?

---

> ### Author Response · Authors · 2025-11-21
> **Response to Reviewer fuiW [1/4]**
>
> We sincerely appreciate your constructive comments and insightful suggestions. Below, we respond to each concern in detail and provide further analysis and clarification.
> > **Q1**: Could you clarify how the latency of each tool call is determined? Is it uniformly set to one round for all tools? If so, what is the justification for this simplification, and do you plan to support heterogeneous or stochastic latency in future versions?
>
> ### **A1**:
> **Delay Setting:** In our main experiment, we uniformly set the tool invocation delay to one turn. This was to minimize noise introduced by random delays during evaluation, making comparisons between different models more controllable and fair. We noted that using two-turn delays or random delays significantly increased the interaction length and the variance of the evaluation results. As shown in Appendices **Tables 8, 9, and 10**, compared to a one-turn delay, two-turn delays and random delays lead to a decrease in overall performance and introduce more randomness, thus reducing the stability of comparisons between different models. For these reasons, we adopted a uniform one-turn delay as the default setting in the main experiment to highlight the impact of delay itself on model inference behavior.
>
> **Future Support:** We sincerely thank the reviewers for their suggestions. It is important to emphasize that our framework already supports more general delay configurations, including heterogeneous delays and random delays; however, in the main experiment, we chose a more stable and controllable scenario as the default setting. In future versions, we are very willing to further expand these experiments, systematically explore the impact of different delay distributions on model behavior, and incorporate them into a more comprehensive evaluation protocol.

---

> ### Author Response · Authors · 2025-11-21
> **Response to Reviewer fuiW [2/4]**
>
> > **Q2**: Have you conducted the experiments without tool-call latency (i.e., synchronous setting)? Such a comparison would help isolate the impact of asynchrony and better demonstrate the value of the benchmark.
>
> ### **A2**:
> **Figure 3** in our paper presents the performance of several models in synchronous settings (Sync TC and Sync SC). The results show that **removing tool invocation latency significantly improves the overall accuracy of the models across almost all tasks**. We further analyze this phenomenon in the discussion section of the paper: in synchronous settings, the models do not need to wait for tool returns, reducing inference interruptions and error accumulation caused by latency, thus resulting in more stable performance.
> We are very grateful to the reviewers for pointing out the importance of this comparison and agreeing that it is key to understanding the challenges of asynchronous scenarios.

---

> ### Author Response · Authors · 2025-11-21
> **Response to Reviewer fuiW [3/4]**
>
> > **Q3**: If a model is strong at single-task tool use, it might achieve high scores by frequently switching tasks with a carefully crafted prompt. Have you observed such behavior? Are there any mechanisms or penalties in place to ensure that the evaluation rewards strategic scheduling rather than heuristic over-invocation?
>
> ### **A3**:
> **Observed Behavior:** Indeed, as the reviewers pointed out, when faced with a "delayed waiting" signal, simply relying on numerous repetitive calls to increase confidence can unnecessarily increase the number of interaction rounds. The examples we present in the paper (such as those in multi-tasking scenarios) illustrate this point: **stronger models tend to proactively switch to other tasks while waiting for a tool to return, thus utilizing time more effectively**; conversely, weaker models tend to repeat calls, significantly increasing the number of dialogue rounds. This is also verified in the statistics of **Table 12**, where smaller models often require more rounds of interaction to complete the same task compared to larger models. We believe that this behavioral difference is a natural manifestation of the difference in "scheduling ability" in asynchronous environments, and is one of the important capabilities that this benchmark aims to characterize.
>
> **Penalty Mechanism:** Currently, our evaluation framework does not directly penalize "excessive calls," but rather reflects this redundant behavior in the overall performance metrics through the increase in the number of interaction rounds. However, we fully agree with the reviewers' approach: in reinforcement learning, designing explicit round-number penalties (e.g., adding R_penalty = −λT during training, where T is the generation round number and λ is the penalty coefficient) can encourage models to be more policy-oriented in scheduling and reduce inefficient repetitive calls. In future model training and framework design, we plan to further explore this direction to better promote the model's learning of truly efficient scheduling strategies.

---

> ### Author Response · Authors · 2025-11-21
> **Response to Reviewer fuiW [4/4]**
>
> > **Q4**: Can you elaborate on the trade-off between evaluation simplicity (using deterministic paths) and realism? Is ASYNCTOOL evaluating planning or just the ability to follow a complex instruction?
>
> ### **A4**:
> **Trade-off between deterministic and diverse execution paths:** Our deterministic path approach primarily targets single-task datasets rather than single multi-task datasets. In current mainstream benchmarks (such as BFCL, Taubennch, and Acebench), the core task of evaluating models is typically to allow the model to select the appropriate tool to complete a given task. In these benchmarks, the task itself is explicit, and the choice of tools is limited; thus, the information and choices differ in each interaction. Our evaluation focuses on **how the model makes its next decision based on the selection of different tools and the results returned by those tools**. Selecting tools and making appropriate decisions based on their feedback is crucial for measuring model capability.
>
> **Diversity of evaluation methods:** To more comprehensively evaluate the performance of individual data points, we do not rely solely on path accuracy. As defined in **Appendix A.1**, we employ multiple validation methods, including validating the execution environment configuration and the executor state output by the model after execution. These metrics provide a more comprehensive examination of model performance, better addressing the diverse execution paths that may be encountered in real-world tasks. Therefore, we strike a balance between simplicity and realism, ensuring that the evaluation is both operable and accurately reflects the model's capabilities. The evaluation goal of AsyncTool: AsyncTool is designed to evaluate a model's time planning capabilities, particularly its ability to efficiently schedule tool calls to complete multiple tasks. The model's ability to handle tool call latency is one of the core aspects we aim to assess. Furthermore, as shown in **Table 3**, the tools we use cover multiple everyday domains. These tools not only meet current open-source tool benchmark standards but are also closely related to common real-world tasks. Therefore, AsyncTool not only evaluates the model's ability to execute complex instructions but, more importantly, examines its overall performance in an asynchronous tool call environment.
>
> We sincerely thank the reviewers for their valuable feedback, which has helped us to more clearly articulate the evaluation framework and design intent of this research.

---

> ### Author Response · Authors · 2025-11-27
> **Looking forward to further discussion**
>
> Dear Reviewer fuiW:
>
> Thank you very much for your suggestions on our paper. We are delighted to receive your positive feedback! Thank you!
>
> Based on your constructive suggestions, we have further improved the paper. Below is a summary of our revisions and analysis in our response:
>
> - We clarified the evaluation objectives of the AsyncTool benchmark and added additional metrics to evaluate the model's utilization of time efficiency.
>
> - We provided a motivational analysis of the delay and more detailed experimental analyses under different experimental settings, along with a comparison of experiments under synchronous conditions.
>
> - We provided solutions to improve model performance on the benchmark and presented the experimental results.
>
> - We updated the figures and improved the paper's wording, and added relevant examples.
>
> We greatly appreciate your valuable time and expertise in reviewing our work. If possible, we would like to ask if, given these updates, it is possible to improve the overall score of this paper? Thank you again for your valuable insights and review!
>
> Thank you very much for your constructive comments, time, and patience.
>
> Sincerely,
>
> Authors

---

### Official Review · Reviewer_9RxN · 2025-10-31

**Soundness:** 3
**Presentation:** 2
**Contribution:** 3
**Rating:** 4
**Confidence:** 3

**Summary:**

This paper introduces ASYNCTOOL, a new benchmark for evaluating LLM agents' capability to handle multiple tasks concurrently through asynchronous tool calls. The authors identify a critical gap in existing benchmarks, which typically assume instantaneous tool responses and focus on single-task scenarios. ASYNCTOOL addresses this by simulating tool response latency, thereby requiring agents to manage idle time by interleaving the execution of different tasks. The paper details the construction of a multi-task dataset via a "hybrid data-evolution strategy" and evaluates a wide range of LLMs using a three-level metric system (Step, Sub-Task, Task). The main finding is that even SOTA models struggle significantly with asynchronous workflows, with the paper providing an analysis of common failure modes.

**Strengths:**

1. Novel Problem Formulation: The paper is the first, to my knowledge, to systematically formalize and create a benchmark for asynchronous multi-task tool use, a critical and under-explored area for LLM agents.
2. Rigorous Data Curation: The multi-step data construction process, combining LLM-based generation with intensive human verification, is a strong point and likely results in a high-quality, internally consistent dataset.
3. Comprehensive Analysis: The paper provides a good qualitative analysis of agent failure modes, offering useful insights into why current models struggle with this task.

**Weaknesses:**

1. Unverifiable SOTA Results: The use of unreleased and hypothetical models (GPT-4.1, GPT-5) for key results is a major flaw. It makes the reported SOTA performance impossible to reproduce and sets an unstable target for the research community. This significantly reduces the benchmark's practical utility.
2. Unrealistic Task Environment: The benchmark's core mechanics—a fixed one-round latency and a single deterministic correct trajectory—do not reflect the variability and flexibility required in real-world scenarios. This limits the generalizability of the findings and risks over-fitting future models to this specific, simplified setup.
3. Lack of Quantitative Error Breakdown: The paper qualitatively describes failure modes but misses the opportunity to provide a quantitative breakdown in the main text. Quantifying the prevalence of "tool confusion" vs. "task neglect" across different models would provide much stronger and more actionable evidence.

**Questions:**

1. Justification for Unreleased Models: Could the authors justify the decision to benchmark and prominently feature hypothetical or private preview models? Given the negative impact on reproducibility and the benchmark's utility, would it not be more scientifically sound to re-frame the results around the best-performing publicly accessible models as the primary baseline?
2. Defense of Deterministic Trajectories: Please provide a stronger defense for enforcing single, deterministic solution paths. Have you analyzed how many tasks in your dataset could have alternative valid paths? How does this design choice not risk penalizing more advanced, flexible reasoning agents in favor of those better at sequence imitation?
3. Impact of the Latency Model: Can you provide any ablation or theoretical argument on how the results might change with a more realistic, variable latency model? Does the current fixed-delay model truly test "temporal reasoning," or does it primarily test context-switching and short-term memory?

---

> ### Author Response · Authors · 2025-11-21
> **Response to Reviewer 9RxN [1/3]**
>
> We sincerely appreciate your constructive
> comments and insightful suggestions. Below, we
> respond to each concern in detail and provide
> further analysis and clarification.
>
> > **Q1**:
> Justification for Unreleased Models: Could the authors justify the decision to benchmark and prominently feature hypothetical or private preview models? Given the negative impact on reproducibility and the benchmark's utility, would it not be more scientifically sound to re-frame the results around the best-performing publicly accessible models as the primary baseline?
>
> ### **A1**:
> Regarding the rationale for simultaneously presenting hypothetical models or proprietary preview models in benchmarks, we would like to provide further explanation here.
>
> Currently, mainstream tool-based benchmarks (such as Taubench, BFCL, and AceBench) report the performance of both closed-source and open-source models in their papers. This practice has become a common approach in the community, and its purpose is not to weaken reproducibility, but to provide researchers with a more comprehensive performance reference, thereby helping to understand the differences in capabilities of different models within a unified framework. **We have also followed this common practice in designing this benchmark.**
>
> We emphasize that the results of closed-source models are for reference only and do not affect the reproducibility of open-source models or the transparency of the evaluation process.**All our experimental setups, tool interfaces, and evaluation pipelines are entirely based on reproducible open-source parts, including the use of repeated experiments for some models and setting the experimental temperature to 0 to ensure reproducibility.** Researchers can reproduce the experimental environment and conduct further research without relying on closed-source models. The results of closed-source models are more for supplementing perspectives, enabling the community to better understand the relative position of open-source models in the current overall model ecosystem, thereby providing a reference coordinate for the future development of open-source models, rather than serving as the sole or core basis for evaluation. We understand the reviewers' concerns regarding scientific principles and strongly agree that open-source models should be kept at the core when reporting benchmarks. In subsequent versions, we are also willing to further strengthen the presentation of the main results for the best open-source model to better highlight its reproducibility.

---

> ### Author Response · Authors · 2025-11-21
> **Response to Reviewer 9RxN [2/3]**
>
> >**Q2**:
> Defense of Deterministic Trajectories: Please provide a stronger defense for enforcing single, deterministic solution paths. Have you analyzed how many tasks in your dataset could have alternative valid paths? How does this design choice not risk penalizing more advanced, flexible reasoning agents in favor of those better at sequence imitation?
>
> ### **A2**:
> **Design of Deterministic Trajectories:** In this study, deterministic trajectories are designed for a single task, not for a single multi-task dataset. Current mainstream benchmarks (such as BFCL, Taubench, and AceBench) generally adopt this approach—the model needs to select an appropriate tool for a given task and make the next decision based on the returned results. The execution trajectory for each task is well-defined, the choice of tools is limited, and the results of each interaction may differ. **Therefore, our core focus is on how the model effectively selects tools and makes decisions based on different tool return values.**
>
> **Consideration of Multiple Paths:** We fully agree with the reviewers' point that multiple effective paths may exist in multi-task datasets. In the paper, we also mention that the model can select different functions to advance task execution based on information such as the current task's return and historical records. Although multiple effective paths exist for each task, our evaluation still ensures that the model can make flexible and effective decisions in a multi-task environment. To avoid simple sequence imitation, we evaluate not only by path accuracy. Appendix A.1 provides various validation methods, including validation of the environment execution configuration and the model's state after execution. These comprehensive evaluation methods ensure a thorough assessment of the model's reasoning capabilities.
>
> **Assessing flexibility and reasoning ability:** Indeed, in each task, the model has multiple effective paths to choose from, and each path may be the optimal solution in some cases. For example, in two different solutions to the same problem, although the results are the same, the path choices are different:
> ```
> step1: get_main_diagonal(matrix=[[1, 2, 3], [4, 5, 6], [7, 8, 9]])
> step2: trim_trailing_dot(string="Hello, world.")
> step3: unary_operator(x=[1, 5, 9])
> step4: find_start_end_indices(input_str="Hello, world", sub_str="world")
> step5: ALL COMPLETED
> ```
> and
> ```
> step1: trim_trailing_dot(string="Hello, world.")
> step2: get_main_diagonal(matrix=[[1, 2, 3], [4, 5, 6], [7, 8, 9]])
> step3: find_start_end_indices(input_str="Hello, world", sub_str="world")
> step4: unary_operator(x=[1, 5, 9])
> step5: ALL COMPLETED
> ```
> We believe that this flexibility and diverse path selection will not penalize more reasoning-capable and flexible agents; on the contrary, it will better reflect the advantages of these models in multi-task asynchronous environments. In such environments, models not only need to handle latency but also need to make reasonable plans based on the real-time state of the task and historical execution records. Therefore, **the evaluation results reflect the comprehensive performance of the model's scheduling and execution capabilities in complex environments. Our benchmark design aims to evaluate these capabilities to ensure that more capable models achieve better scores in asynchronous and multi-task scenarios, which is the core goal of our benchmark design.**
>
> Therefore, the design of deterministic trajectories will not favor models with strong sequence imitation; on the contrary, it helps to examine more flexible and reasoning-capable models and ensures that their performance in asynchronous tool invocation environments is reasonably evaluated.

---

> ### Author Response · Authors · 2025-11-21
> **Response to Reviewer 9RxN [3/3]**
>
> > **Q3**: Impact of the Latency Model: Can you provide any ablation or theoretical argument on how the results might change with a more realistic, variable latency model? Does the current fixed-delay model truly test "temporal reasoning," or does it primarily test context-switching and short-term memory?
>
> ### **A3**:
> **Ablation Experiments and Theoretical Validation:** To better observe the impact of latency on model performance, we present experimental results under different latency settings in **Tables 8, 9, and 10** of the appendix, including configurations with a latency of two rounds, zero to one random round, and one to two random rounds. By comparing the results in **Tables 2 and 8**, we found that overall performance decreased as latency increased. We speculate that increased latency leads to longer interaction rounds in the model, thus increasing the difficulty and complexity of the task. In **Tables 9 and 10**, we observed that random latency introduced significant fluctuations in results. This randomness introduces a certain degree of unfairness; therefore, we consider these results as additional experimental references rather than benchmark evaluation metrics.
>
> **Examination of "Temporal Reasoning Ability":** We understand the reviewers' concerns about whether fixed latency truly tests "temporal reasoning ability." In current mainstream benchmarks (such as BFCL, Taubench, and AceBench), a specific task is usually given, requiring the model to choose a tool to complete the task. The task itself is deterministic, the tool selection is limited, and the information and choices brought by each interaction are different. Therefore, the core of the evaluation is how the model selects tools and makes appropriate decisions based on the returned results. We introduced a delay setting in information selection, which places higher demands on the model's real-time planning and decision-making, especially in multi-tasking environments where the model needs to rationally schedule tool calls.
>
> **Temporal Reasoning and Decision-Making Capabilities in Multi-Task Asynchronous Environments:** In the AsyncTool evaluation, we focus more on examining the model's time planning capabilities. In a multi-task asynchronous environment, the model faces multiple tasks to be completed, and each task's tool call has a delay. Under this setting, the model cannot simply rely on continuously advancing a single task, but needs to rationally adjust the task advancement order based on the current task status and historical records. This process examines the model's temporal understanding ability, i.e., how the model manages and schedules tasks in an asynchronous and delayed environment.
>
> **Short-Term Memory and Decision-Making Capabilities:** Closely related to temporal reasoning ability are the model's short-term memory and decision-making capabilities. When faced with multiple tasks, the model needs to combine the historical trajectory of each task to decide how to proceed to the next operation. Therefore, the evaluation not only tests the model's temporal understanding but also examines its decision-making ability in the passage of time and the effective use of short-term memory.
>
> In summary, our benchmark tests aim to comprehensively evaluate the model's performance in multi-task, asynchronous and delayed environments. By introducing delays and task switching, the evaluation not only covered temporal reasoning capabilities but also tested the model's short-term memory, task scheduling, and flexible decision-making abilities. Therefore, even with a more realistic variable-delay model, we were still able to effectively evaluate the model's capabilities in complex reasoning and task planning.
>
> We thank the reviewers for their insightful attention to this issue and hope our explanation clarifies our design considerations regarding the delay settings.

---

> ### Author Response · Authors · 2025-11-27
> **Looking forward to futher discussion**
>
> Dear Reviewer 9RxN:
>
> Thank you very much for your suggestions on our paper. We are delighted to receive your positive feedback! Thank you!
>
> Based on your constructive suggestions, we have further improved the paper. Below is a summary of our revisions and analysis in our response:
>
> - We clarified the evaluation objectives of the AsyncTool benchmark and added additional metrics to evaluate the model's utilization of time efficiency.
>
> - We provided a motivational analysis of the delay and more detailed experimental analyses under different experimental settings, along with a comparison of experiments under synchronous conditions.
>
> - We provided solutions to improve model performance on the benchmark and presented the experimental results.
>
> - We updated the figures and improved the paper's presentation, and added relevant examples.
>
> We greatly appreciate your valuable time and expertise in reviewing our work. If possible, we would like to ask if, given these updates, it is possible to improve the overall score of this paper? Thank you again for your valuable insights and review!
>
> Thank you very much for your constructive comments, time, and patience.
>
> Sincerely,
>
> Authors

---

### Author Response · Authors · 2025-11-26

We thank the reviewers and the domain leader for reviewing our manuscript. We have responded to all reviewers' comments and clarified our contributions based on their valuable suggestions. To address the reviewers' concerns, we conducted several additional experiments and analyses to verify the effectiveness and innovativeness of AsyncTool, and added 3 pages of content to the revised manuscript, summarized as follows:

- We clarified that deterministic trajectories are only applicable to single tasks, while multi-task scenarios inherently possess multiple efficient paths, and supplemented with relevant data examples.

- We clarified that our evaluation method can accurately identify different trajectories, and added the model's average rounds graph as a specific scheduling metric.

- We provided a motivational analysis of the latency and supplementary experiments under different latency levels (Tables 8/9/10), along with experimental comparisons under synchronous conditions.

- We supplement our solution regarding asynchronous tool calls and present the experimental results of the model under the few-shot setting (Table 11).

- We clarify the evaluation objectives of the AsyncTool benchmark and conduct a comparative analysis with mainstream benchmarks (BFCL, taubench, etc.).

- We provided detailed answers and experimental analyses to the reviewers' questions, and updated the graphs and improved the paper's presentation based on their suggestions.

If you still have any questions about this work, please feel free to contact us. We would be happy to discuss it with you and conduct further experiments on any evidence that may interest you.

---

### Meta-Review · Area_Chair_vUQC · 2026-01-02

**Summary:**

The submission proposes ASYNCTOOL, a benchmark for evaluating LLM agents in asynchronous, multi-task tool-use scenarios with delayed tool responses.
Reviewers generally agree that the paper addresses a timely and underexplored problem and that the benchmark is well-motivated and carefully constructed. However, there is also broad consensus that the current benchmark design involves significant simplifications (fixed latency, deterministic trajectories, correctness-only metrics) that limit realism and generalizability, resulting in mixed review scores.

The most consistent concern is the simplified latency model (all major reviewers). The benchmark relies on a fixed one-round tool delay, which reviewers felt does not adequately reflect the stochastic and heterogeneous nature of real-world tool latencies. Although the authors provide ablation studies with variable delays in the appendix, these are not integrated into the core benchmark, leading reviewers to question whether the evaluation primarily measures context switching and short-term memory rather than genuine temporal reasoning.

A second major concern is the use of deterministic reference trajectories. Reviewers worried that enforcing a fixed execution path risks turning the task into plan following rather than plan generation. While the authors have clarified that evaluation is not strictly path-based, reviewers remained cautious about the trade-off between evaluation simplicity and realistic planning.

Reviewers also consistently noted the lack of explicit efficiency or scheduling metrics. The evaluation focuses largely on correctness, without directly measuring idle-time utilization, scheduling optimality, or tool-call efficiency.

Additional concerns include the use of unreleased or hypothetical models. which harms reproducibility and practical benchmarking value, and several presentation clarity issues (e.g., table interpretability and explanation of latency injection)

Overall, reviewers viewed the paper as a promising but incomplete first step, with strong motivation and analysis but important methodological simplifications that prevent it from being a fully convincing or mature benchmark at this stage.

**Reviewer Concerns:**

Reviewer concerns that have been addressed by the rebuttal:
* Fixed one-round latency versus realistic variable latency
* Use of unreleased or hypothetical models, regarding reproducibility concern
* Missing comparison to synchronous (no-latency) setting
* Presentation and clarity issues

The review concerns that are still outstanding:
* Deterministic trajectories and "plan following" versus "plan generation"
* Lack of explicit scheduling or efficiency metrics
* Risk of heuristic over-switching or tool overuse (Reviewer fuiW concerns that models may game the benchmark via excessive task switching or repeated calls)
* Lack of explored solutions or mitigation strategies

**Reviewer Scores:**

Reviewer 9RxN (remains 4 weak rejection): This reviewer was strongly critical of unreleased models and deterministic trajectories, and fixed latency.

Reviewer fuiW (remains 6 weak acceptance): This reviewer already leaned slightly positive but expressed reservations about realism, efficiency metrics, and deterministic paths.

Reviewer adPW (remains 4 weak rejection): Although the authors have already addressed some of this reviewer's concerns, there are some remaining issues like efficiency metrics and lack of mitigation strategies.

---

### Decision · Program_Chairs · 2026-01-26

Reject